🔓 | Microbial Pathogenesis | Research Article

# A highly conserved tRNA modification contributes to *C. albicans* filamentation and virulence

Bettina Böttcher,[1,2] Sandra D. Kienast,[3,4] Johannes Leufken,[3,4] Cristian Eggers,[3,4] Puneet Sharma,[3,4] Christine M. Leufken,[3] Bianka Morgner,[1] Hannes C. A. Drexler,[5] Daniela Schulz,[1] Stefanie Allert,[1] Ilse D. Jacobsen,[6,7] Slavena Vylkova,[2] Sebastian A. Leidel,[3,4] Sascha Brunke[1]

**ABSTRACT** tRNA modifications play important roles in maintaining translation accuracy in all domains of life. Disruptions in the tRNA modification machinery, especially of the anticodon stem loop, can be lethal for many bacteria and lead to a broad range of phenotypes in baker's yeast. Very little is known about the function of tRNA modifications in host-pathogen interactions, where rapidly changing environments and stresses require fast adaptations. We found that two closely related fungal pathogens of humans, the highly pathogenic *Candida albicans* and its much less pathogenic sister species, *Candida dubliniensis*, differ in the function of a tRNA-modifying enzyme. This enzyme, Hma1, exhibits species-specific effects on the ability of the two fungi to grow in the hypha morphology, which is central to their virulence potential. We show that Hma1 has tRNA-threonylcarbamoyladenosine dehydratase activity, and its deletion alters ribosome occupancy, especially at 37°C—the body temperature of the human host. A *C. albicans HMA1* deletion mutant also shows defects in adhesion to and invasion into human epithelial cells and shows reduced virulence in a fungal infection model. This links tRNA modifications to host-induced filamentation and virulence of one of the most important fungal pathogens of humans.

**IMPORTANCE** Fungal infections are on the rise worldwide, and their global burden on human life and health is frequently underestimated. Among them, the human commensal and opportunistic pathogen, *Candida albicans,* is one of the major causative agents of severe infections. Its virulence is closely linked to its ability to change morphologies from yeasts to hyphae. Here, this ability is linked—to our knowledge for the first time—to modifications of tRNA and translational efficiency. One tRNA-modifying enzyme, Hma1, plays a specific role in *C. albicans* and its ability to invade the host. This adds a so-far unknown layer of regulation to the fungal virulence program and offers new potential therapeutic targets to fight fungal infections.

**KEYWORDS** tRNA modification, *Candida albicans*, host-pathogen interactions, virulence regulation

Address correspondence to Sascha Brunke, sascha.brunke@leibniz.hki.de.

The authors declare no conflict of interest.

See the funding table on p. 22.

tRNA molecules present the structural interface between the RNA and protein worlds. In addition to their decoding function, tRNAs play important roles in translation accuracy and dynamics, and in primary metabolism (1) as well as stress responses (2). These functions depend on a plethora of post-transcriptional chemical modifications of the tRNA molecule itself. For example, a typical baker's yeast exhibits roughly 25 different types of modifications on 36 different locations in its cytoplasmatic tRNA molecules (3), thus making tRNAs the most heavily modified of all RNA species. The anticodon loop is a hot spot of modifications that are critical for translational efficiency and fidelity (4). The purine base 37 is frequently modified by a variety of chemical groups (5). tRNA

modifications are introduced by a diverse group of enzymes. Prominent examples are the elongator complex (6) and the *URM1* pathway (7), which catalyze wobble uridine modifications (mcm$^5$s$^2$U$_{34}$) and the threonyl-carbamoyl transferase complex, comprising Tcs1-Tcs7 in *Saccharomyces cerevisiae* (8, 9), for adenosine $N^6$-threonylcarbamoyladenosine (t$^6$A$_{37}$) modifications at the anticodon stem loop (ASL). Most common and well conserved are t$^6$A, $N^6$-isopentenyladenosine (i$^6$A), and 1-methylguanosine (m$^1$G), which facilitate mRNA-to-tRNA binding and prevent misreading. Furthermore, translation-frame selection is affected by t$^6$A during decoding of the start codon (1, 10, 11). t$^6$A, first described in 1974 (12), was later shown to be a hydrolyzation product of cyclic threonyl-carbamoyladenosine (ct$^6$A), which is commonly found in bacteria, fungi, and plants (13). Defects in modifications of the ASL are lethal for most bacteria (14) and result in a broad range of phenotypes in yeast, including increased susceptibility to diverse stressors (15), defects in translation, and aggregation of misfolded proteins (9, 16).

Finally, tRNAs act as signaling molecules for the nutritional state of cells. Uncharged tRNAs accumulate under amino acid depletion and are recognized by the Gcn2 kinase (17). Gcn2 subsequently activates transcription factors for the expression of amino acid biosynthesis genes. This Tor1 kinase-dependent pathway is known as the general amino acid control and has been associated with tRNA modifications of the ASL: A *S. cerevisiae tcs2Δ* mutant, defective in t$^6$A modification of ANN codon-tRNAs, lacks Gcn2-dependent Gcn4 activation (18), and cells that miss members of the *URM1* pathway are sensitive to rapamycin (19). As yeast requires thiolated tRNAs to balance metabolic carbon and amino acid fluxes (20), its nitrogen starvation-induced pseudohyphal growth is blocked by a lack of tRNA thiolation at the ASL (21).

*Candida* species are fungal pathogens which frequently cause bloodstream infections with high mortality rates. For these (often) nosocomial infections, *Candida albicans* is by far the leading cause (22, 23). In contrast, its closest relative, *Candida dubliniensis*, rarely causes life-threatening infections (24), even though it is detectable in up to one-third of oral candidosis in HIV-infected patients (25). Although the vast majority of *C. dubliniensis* genes share sequence identities of over 80% with their *C. albicans* orthologs, certain pathogenicity-related genes differ markedly in sequence or are even absent in *C. dubliniensis* (26). These genomic differences are reflected in *in vitro* conditions, where *C. dubliniensis* is less robust against oxidative, osmotic, and temperature stresses (27).

Both *C. albicans* and *C. dubliniensis* are polymorphic fungi that rely on yeast-to-hyphae transitions for tissue invasion and dissemination within their host (25, 28). However, the induction of hyphae by nutrient availability differs significantly between the two species, implying differences in Tor1-dependent regulation (29). Furthermore, on reconstituted human epithelium, *C. dubliniensis* fails to produce germ tubes and cannot invade the tissue—in stark contrast to the highly invasive *C. albicans*. The overall lower virulence of *C. dubliniensis* also manifests in animal models like mice (30) or embryonated chicken eggs (31). Therefore, investigating the genetic differences between *C. albicans* and *C. dubliniensis* is an excellent model to gain key insights into species-specific stress responses, regulators of morphogenesis, and pathogenicity-related pathways in human infections.

In this study, a cross-species approach reveals a novel role for the *C. albicans* tRNA-modifying enzyme, Hma1, during starvation-induced morphogenesis and links it to the TOR pathway. We confirm the molecular function of Hma1 by quantifying t$^6$A levels as a proxy for Hma1 activity and determine its role in translation by codon-specific ribosome-profiling analyses. Finally, we show the strong impact of *HMA1* on *C. albicans* pathogenesis *in vitro* and *in vivo*.

## MATERIALS AND METHODS

### Reagents

Antibodies were acquired from Acris (rabbit α-*Candida*, #BP1006) and Abcam (goat α-rabbit, Alexa Fluor 488-coupled, #ab150077). Kits were bought from GE Healthcare (ECL Direct Nucleic Acid Labeling and Detection Kit, #RPN3001) and Takara Bio Europe (InFusion Cloning Kit, #638916). Specific chemicals were obtained from Sigma-Aldrich, Germany: antimycin A (#A8674), caffeine (#C0750), calcofluor white (#910090), cyclohex-imide (#01810), L-homoserine (#8.14359), menadione (#M5625), oligomycin (#O4876), and rapamycin (#R8781). Nourseothricin was bought from Werner Bioagents, Germany (#5.010.000). The AXIO Observer.Z1 was bought from Carl Zeiss, Germanyl.

### Biological resources

All *Candida albicans* and *Candida dubliniensis* strains in this work are listed in Table 1. The TR146 cell line was originally obtained from the European Collection of Cell Cultures as lot number 13D016. Identity of the cell line and absence of mycoplasm contamination were tested regularly before, during, and after the experiments described here. Fertilized chicken eggs (breed "weiße Leghorn") were obtained from local producers (32).

**TABLE 1** *Candida* strains used in this study

| Strain | Parent | Genotype | Reference |
|---|---|---|---|
| SC5314 | *C. albicans* wild-type strain | | (33) |
| CaCSP1-GFP | SC5314 | orf19.3512/orf19.3512-GFP-T$_{ACT1}$ | (34) |
| *HMA1/SAT1* A/B | SC5314 | orf19.2115Δ::*SAT1*-FLIP/orf19.2115 | This study |
| *HMA1/hma1Δ* A/B | *HMA1/SAT1* A/B | orf19.2115Δ::FRT/orf19.2115 | This study |
| *hma1Δ/SAT1* A/B | *HMA1/hma1Δ* A/B | orf19.2115Δ::FRT/orf19.2115Δ::*SAT1*-FLIP | This study |
| *hma1Δ/Δ* A/B | *hma1Δ/SAT1* A/B | orf19.2115Δ::FRT/orf19.2115Δ::FRT | This study |
| *HMA1*KS1 | *hma1Δ/Δ*B | orf19.2115Δ::FRT/orf19.2115Δ::orf19.2115-T$_{ACT1}$-*SAT1*-FLIP | This study |
| *hma1Δ/HMA1* | *HMA1*KS1 | orf19.2115Δ::FRT/orf19.2115Δ::*orf*19.2115-T$_{ACT1}$-FRT | This study |
| Wü284 | *C. dubliniensis* wild-type strain | | (35) |
| Cd*CSP1*-GFP | Wü284 | CD36_30750/CD36_30750-GFP-T$_{ACT1}$ | (34) |
| CdUM4A | Wü284 | *Cdura3Δ*::FRT/*Cdura3Δ*::FRT | (36) |
| CdINT1 | CdUM4A | *Cdura3Δ*::FRT/*Cdura3*::pCdINT1 | (37) |
| CdINT4 | CdUM4A | *Cdura3Δ*::FRT/*Cdura3*::pCdINT1-Ca21chr2:22964-27892W | This study |
| Cd2113 | CdUM4A | *Cdura3Δ*::FRT/*Cdura3Δ*::pCdINT1-orf19.2113 | This study |
| Cd2114 | CdUM4A | *Cdura3Δ*::FRT/*Cdura3Δ*::pCdINT1-orf19.2114 | This study |
| Cd2115 | CdUM4A | *Cdura3Δ*::FRT/*Cdura3Δ*::pCdINT1-orf19.2115 | This study |
| Cd*HMA1/SAT1* A/B | Wü284 | CD36_15150Δ::*SAT1*-FLIP/CD36_15150 | This study |
| Cd*HMA1/Cdhma1Δ* A/B | Cd*HMA1/SAT1* A/B /B | CD36_15150Δ::FRT/CD36_15150 | This study |
| Cd*hma1Δ/SAT1* A/B | Cd*HMA1/Cdhma1Δ* A/B | CD36_15150Δ::FRT/CD36_15150Δ::*SAT1*-FLIP | This study |
| Cd*hma1Δ/Δ* A/B | Cd*hma1Δ*/SAT1 A/B | CD36_15150Δ::FRT/CD36_15150Δ::FRT | This study |
| Cd*HMA1*KS1 | Cd*hma1Δ/Δ* | CD36_15150Δ::FRT/CD36_15150Δ::CD36_15150-T$_{ACT1}$-*SAT1*-FLIP | This study |
| *Cdhma1Δ/HMA1* | Cd*HMA1*KS1 | CD36_15150Δ::FRT/CD36_15150Δ::CD36_15150-T$_{ACT1}$-FRT | This study |
| Cd2115_Cd*HMA1/SAT1* A/B | Cd2115 | *Cdura3Δ*::FRT/*Cdura3Δ*:: pCdINT1-orf19.2115 CD36_15150Δ::*SAT1*-FLIP/CD36_15150 | This study |
| Cd2115_Cd*HMA1/Cdhma1Δ* A/B | Cd*HMA1*M1A/B | *Cdura3Δ*::FRT/*Cdura3Δ*:: pCdINT1-orf19.2115 CD36_15150Δ::FRT/CD36_15150 | This study |
| Cd2115_Cd A/B | Cd*HMA1*M2A/B | *Cdura3Δ*::FRT/*Cdura3Δ*::pCdINT1-orf19.2115 CD36_15150Δ::FRT/CD36_15150Δ::*SAT1*-FLIP | This study |
| Cd2115_Cd*hma1Δ/Δ* A/B | Cd*HMA1*M3A/B | *Cdura3Δ*::FRT/*Cdura3Δ*:: pCdINT1-orf19.2115 CD36_15150Δ::FRT/CD36_15150Δ::FRT | This study |

## Statistical analyses

For all experiments, $P < 0.05$ was considered statistically significant. Lower $P$-values are indicated by asterisks, with *, $P < 0.05$; **, $P \leq 0.01$; ***, $P \leq 0.001$. For data with expected normal distribution, two-sided $t$-tests were used with a biological replicate count $n \geq 3$ (indicated in the figure legends). All data are shown as mean with standard deviation of biological (independent) replicates. In case of multiple testing, an appropriate correction was used (Benjamini-Hochberg or Šidák as indicated). For the infection experiments, the Mantel-Cox test was used to detect statistical significance of the difference between survival curves, with $n = 20$ for each group.

## Culture conditions

*Candida* strains were routinely propagated on YPD agar (20 g peptone, 10 g yeast extract, 20 g glucose, 15 g agar per liter) at 30°C and stored as frozen stocks in YPD medium with 15% (vol/vol) glycerol at −80°C.

## Morphological tests

### Chlamydospore formation

Chlamydospore production by *C. dubliniensis* was induced on solid synthetic low-ammonium dextrose (SLAD) agar (0.17% Yeast Nitrogen Base without Amino Acids and Ammonium Sulfate; BD, Heidelberg, Germany) (38) without any additional nitrogen source but with 2% glucose and 2% agar (Kobe I, Carl Roth GmbH + Co. KG, Germany). Formation of chlamydospores by *C. albicans* was induced on corn meal-Tween 80 agar (BD, Heidelberg, Germany). The plates were incubated at 28°C for 2–7 days in darkness, and chlamydospore formation was monitored microscopically (Axiovert, Zeiss, Germany).

### Germ tube assays

For hyphae induction in liquid media, *Candida* strains were pre-grown in liquid YPD overnight (30°C, 180 rpm), washed with phosphate-buffered saline (PBS), and $10^5$ cells were transferred into 500 μL + $H_2O$ plus 10% (vol/vol) fetal calf serum (FCS) and the morphology was analyzed microscopically (Axiovert, Zeiss, Germany). *ECE1* transcript levels were determined using the same medium. RNA was isolated from yeasts (0 h) and hyphae (6 h) using a glass bead method with the Qiagen RNeasy Kit. RNA quality and quantity were measured using an Agilent BioAnalyzer and a Nanodrop instrument, respectively. cDNA was synthesized using Invitrogen Superscript III following the protocol from reference (39).

### Filamentation assay

*Candida* cells from YPD overnight cultures were washed, and 5 μL of $2 \times 10^7$ cells/mL was spotted on the following agar plates: YPD, Spider, water agar, SLAD, or boiled blood (chocolate) agar (40). Colony filamentation was investigated after incubation at 37°C in either atmospheric or 5% $CO_2$-enriched air after 5 days incubation, using a binocular (Stemi 2000-C, Zeiss, Germany).

## Stress assays

Resistance of *Candida* strains to stressors was tested by growth curve assays or by serial drop dilution tests on solid agar plates. In both cases, *Candida* cells from YPD overnight cultures (30°C, 180 rpm) were initially washed twice with water. For growth in liquid medium, cells were diluted to $OD_{600} = 0.01$ in YPD or in synthetic minimal media: SD (Yeast Nitrogen Base with Ammonium Sulfate, MP, Santa Ana, CA, USA) or YCB medium (Yeast Carbon Base, BD, Heidelberg, Germany).

Stress tests were conducted with indicated concentrations of supplements. YPD medium contained 5 or 7.5 nM rapamycin, 6 or 8 mM caffeine, 2.5 or 5 mg/mL

cycloheximide, 100 mM menadione, 2 µM antimycin A, 2 µM oligomycin, 1 µg/mL calcofluor white, and 1M NaCl. YCB medium contained 1 mg/mL L-homoserine and 5 mg/mL ammonium sulfate; SD medium was used as growth control (All stressors Sigma-Aldrich GmbH, Germany). Growth cultures were incubated at 30°C was determined after 30-s shaking every 20 min over 65 h.

For drop dilution tests on solid agar plate, *Candida* cultures were adjusted to $2 \times 10^8$ cells/mL and successively diluted by factor 10. A volume of 5 µL of each concentration was spotted onto solid agar plates, and colony growth was followed for 2 or 3 days under the indicated conditions. Furthermore, 2% glucose was used as standard carbon source of YP (20 g peptone, 10 g yeast extract, and 15 g agar per liter) plates and was replaced with 2% glycerol or ethanol (Sigma-Aldrich GmbH, Germany) to induce respiratory growth. Images were taken with an imaging system (Vilber-Lourmat GmbH, Germany).

## Adhesion/invasion assay

The capability of *C. albicans* cells to adhere and invade host cell layers was assayed by infecting monolayers of human oral epithelial TR146. Cell cultivation, infection, and fixation procedures were performed as previously described (41) using an MOI of 0.4. Three hours post infection, non-adherent cells were rinsed with PBS, and samples were fixed with Histofix (Carl Roth GmbH + Co. KG, Germany). Adherent *Candida* cells on the epithelium were marked with primary anti-*Candida* antibody (Acris Anti *Candida*, rabbit*,* Herford, Germany) and detected with an Alexa Fluor 488-coupled secondary anti-rabbit antibody. Subsequent permeabilization of the TR146 cells with 0.5% Triton X-100 allowed staining of invasive hyphae parts with calcofluor white (Sigma-Aldrich GmbH, Germany). Hyphal length and invasiveness were determined for 100 hyphae of each strain in triplicates using fluorescence microscopy (Zen2 pro, Zeiss, Germany).

## Screening of a *C. albicans* genomic library and identification of the *C. albicans* genomic fragments inserted in *C. dubliniensis*

The construction of a genomic *C. albicans* library in *C. dubliniensis* was described before by Staib and Morschhäuser (37). Approximately 20,000 library clones were screened on SLAD agar under chlamydospore-inducing conditions, and colonies with altered colony morphology were confirmed on SLAD agar. The *C. albicans*-derived DNA fragment from the transformant CdINT4 was amplified and sequenced using the primers CdUra14 and M13rev. Sequence analysis demonstrated that CdINT4 contained *C. albicans* sequences corresponding to chromosomal coordinates 22964–27892 of chromosome 2. Three complete *C. albicans* ORFs, orf19.2113, orf19.2114, and orf19.2115, are located within this region. Each ORF, including native promotor and terminator regions, was amplified via PCR using the corresponding primer pairs (IF2113_1/2, IF2114_1/2, IF2115_1/2) and cloned into *ApaI/XhoI*-cut plasmid pcdINT1 using the InFusion Cloning Kit (Takara Bio Company). The resulting plasmids were transformed (see *Candida* transformant construction) into *C. dubliniensis* CdUM4A (36), and their morphology was investigated.

## Plasmid construction

The deletion cassette for orf19.2115 was constructed as follows: An *ApaI-XhoI* fragment with orf19.2115 upstream sequences was cloned after amplification by PCR with the primers 2115_3 and 2115_4 (Table S1) using genomic DNA from *C. albicans* SC5314 as template. A *SacII-SacI* fragment containing orf19.2115 downstream sequences was obtained with the primers 2115_5 and 2115_6. These orf19.2115 upstream and downstream fragments were used to replace *SSU2* upstream and downstream fragments in plasmid pSSU2M2 (42) via the introduced restriction sites, to result in p2115M2, in which the *SAT1* flipper cassette is flanked by orf19.2115 sequences. The gene deletion cassette pcd15150M2 for CD36_15150 in *C. dubliniensis* was constructed analogously using the primer pairs 15150_3 and 15150_4 or 15150_5 and 15150_6, respectively, with gDNA from *C. dubliniensis* Wü284 as template.

The orf19.2115 gene was amplified for genetic reconstitution in *C. albicans* using the primers 2115_3 and 2115_7. The *ApaI/BamHI*-cut DNA fragment was integrated into pSAP2KS1 (43) resulting in p2115KS1. The orf19.2115 downstream DNA fragment was amplified using the primers 2115_8 and 2115_9, and the *NotI/SacII*-cut DNA fragment was ligated into p2115KS1. For the CD36_15150 revertant in *C. dubliniensis,* the whole CD36_15150 gene was amplified using the primer pair 15150_3 and 15150_7, and the downstream DNA fragment using the primers 15150_5 and 15150_6 c. The *ApaI/SmaI*- or *SacII/SacI*-cut fragments were integrated into pSAP2KS1.

### *Candida* transformant construction

Chemically competent *C. albicans* or *C. dubliniensis* cells were transformed with linear DNA fragments by heat shock at 44°C (44, 45), and resulting clones were selected on YPD plates containing 50–200 μg/mL nourseothricin (Werner Bioagents, Jena, Germany). The *SAT1* flipper strategy allows the recycling of the selection marker (46). The insertion locus of the DNA fragment was confirmed by Southern Blot analyses.

### Southern blot

Ten micrograms of isolated genomic DNA were digested with an appropriate restriction enzyme. After DNA separation on an agarose gel (1%), DNA was transferred onto a nylon membrane using a vacuum blot system. UV-linked DNA was hybridized with chemiluminescence-enabled probes and detected via the Amersham ECL Direct Nucleic Acid Labeling and Detection (GE Healthcare, Braunschweig, Germany) according to the manufacturer's instructions.

### Reversed phase nano liquid chromatography mass spectrometry of ribonucleosides

*Candida* strains were grown overnight in liquid YPD cultures (30°C, 180 rpm), washed, and diluted to an $OD_{600}$ of 0.2 in 50 mL YPD medium. Cultures were grown for additional 4 h at 30°C or 37°C. Cells were harvested by centrifugation, frozen in liquid nitrogen, and stored at −80°C until lysis. Total RNA was isolated according to the protocol by Miyauchi et al. (13) using acidic phenol-TRIzol (Life Technologies GmbH, Darmstadt, Germany) extraction. The samples were kept on ice whenever possible. The tRNA was subsequently isolated by gel extraction from denaturing 8 M urea 8% polyacrylamide gels (47). For each biological replicate, we prepared two to three technical replicates depending on the amount of tRNA that we extracted from the gel (50 samples in total for *C. albicans* wt and revertant, 50 for *C. albicans* deletion mutants, and 27 for *C. dubliniensis* wt; Table S2). The gel-extracted tRNAs were enzymatically digested under acidic conditions and dephosphorylation into single ribonucleosides as described (48). The digested ribonucleosides were dissolved in 5 mM $NH_4HCO_2$, pH 4.8, spiked with $^{15}N$-labeled ribonucleosides from *Chlamydomonas reinhardtii,* and subsequently subjected to reversed phase chromatography using a self-packed C18 (Synergi 4 μm Hydro-RP 80 Å; Phenomenex Ltd., Aschaffenburg, Germany) capillary column (75 μm ID × 430 mm) coupled to a Proxeon EASY nLC (Thermo Fisher Scientific GmbH, Dreieich, Germany). A multi-step gradient (0%–38% B in 31 min, 38%–50% B in 6 min, 50%–100% B in 6 min, hold at 100% B for 16 min) with a solvent system consisting of $NH_4HCO_2$, pH 4.8 (solvent A) and 40% acetonitrile (solvent B), and a flow rate of 250 nL/min was applied. On-column temperature was set to 20°C. Electrospray ionization mass spectrometry analysis was performed using a Q Exactive mass spectrometer (Thermo Finnigan LLC, San Jose, CA) operated in the positive mode at a resolution of 70,000, the AGC (automated gain control) target value set to $3 \times 10^6$, and the fill time to 50 ms. Full MS spectra (m/z 100–700) and Top5 ddMS2 spectra were recorded. Immediately before or after the measurements of the biological samples, we measured chemical standards of $t^6A$ and $ct^6A$ to determine their elution time from the gradient. The identity of the chemical standards was verified by MS2 fragmentation.

## Quantitative analysis of LC-MS/MS data

Acquired Thermo RAW files were converted to mzML format (49) using msconvert as part of Proteowizard (version 3.0.11799) (50). Data analysis was carried out using MORAST (Leufken et al., unpublished data) using default parameters, and the fragmentation method set to "stepped_low_hcd." Within MORAST, spectra were accessed and centroided, if necessary, by pymzML (version 2.0.2) (51). Furthermore, pyQms (version 0.5.0) was used for quantification of all known ribonucleosides (52) [data obtained from MODOMICS; (53)], in MS1 (5 ppm accuracy) and MS2 (20 ppm accuracy) spectra. Metabolic labels were defined for nitrogen using either an enrichment of 0% (unlabeled) or 99% (fully labeled) of $^{15}$N. LC-MS/MS runs were corrected for systematic m/z errors, if necessary. Spiked-in $^{15}$N-labeled nucleosides were used for internal quality control and normalization. The identities of quantified ribonucleosides were verified by their specific fragmentation patterns in MS2 and by predetermined chromatographic elution orders of structural isomers (48). Abundances for all ribonucleosides were determined using the area under curve for each chromatographic peak. Furthermore, MORAST uses the Python packages: SciPy (54), Scikit-learn (55), and statsmodels (56). pyQms (57) calculates an accurate isotope pattern for each molecule and generates a quality score for each MS1 measurement. Therefore, it does not only use the m/z value of a given molecule but also extracts the chemical sum formula of each ion. The ct$^6$A signal in the biological measurements was so low that the mass spectrometer did not select the ion for MS2 fragmentation. Therefore, we report the detection of a weak ct$^6$A signal in cases where pyQms detected an ion with the sum formula of ct$^6$A ($C_{15}H_{18}N_6O_7$) and a high pyQms quality score at the same retention time as the ct$^6$A chemical standard.

## Ribosome profiling

Ribosome profiling was conducted essentially as described (47, 58, 59). *C. albicans* and *C. dubliniensis* cells were grown to OD$_{600}$≈ 0.4, harvested by vacuum filtration using a 0.45-µm cellulose nitrate filter (GE Healthcare), and flash frozen in liquid nitrogen. The cells were lyzed in a Freezer-Mill (SPEX SamplePrep) using two cycles at 5 CPS and an intermittent cooling step of 2 min. The lysates were thawed in lysis buffer (20 mM Tris-HCl pH 7.4, 5 mM MgCl$_2$, 100 mM NaCl, 1% Triton, 2 mM DTT, 100 µg/mL cycloheximide) and clarified by two rounds of centrifugation (5 min; 4°C; 10,000 g). And, 10 A$_{260}$ units of the lysates were digested for 1 h at 22°C using 600 U RNase I (Ambion, Thermo Fisher). Subsequently, the reaction was stopped by 15 µL SuperaseIn (Thermo Fisher), monosomes were isolated, and 27–29-nt-long footprints were excised from an acrylamide gel. During library generation, 3′-adapters introduced four randomized positions to reduce ligation biases (47). Following sequencing, the adapter sequences were clipped, and the four randomized nucleotides removed using the FASTX-Toolkit. The processed reads were mapped to ORFs (cgdGene) using bowtie (60). Reference ORFs (C_albicans_SC5314_version_A22, allele A) were extended by 18 nt into the UTRs. A-site codons were mapped according to the frame of the 5′ end of footprints—excluding reads that mapped to the first or last 15 codons of each transcript—and an appropriate offset was defined (16). Differential gene expression analysis was performed using DESeq2 (61). For altered transcripts, the *P* values were adjusted with Benjamini-Hochberg correction, and a-threshold was set to 0.05. To identify unaltered transcripts, the althypothesis function was set to "lessAbs."

## Gene ontology analysis

Significant differences in gene expression of the *C. albicans hma1Δ/Δ* mutant (cutoff $P_{adj}$ <0.05 and log$_2$FC ± 1) were analyzed for enriched biological process using the CGD gene ontology (GO) term finder online tool with default settings [*Candida* Genome Database http://www.candidagenome.org/ (62) access in April 2021] and summarized by removing redundant terms using REVIGO (63). Enrichment of regulated GO terms was calculated as the ratio of cluster frequency to background frequency.

## Chicken embryo infection model

The embryonated chicken infection model was used to study virulence as described previously (32). Briefly, overnight cultures of yeasts were washed with PBS and adjusted to $10^8$ or $10^6$ cells/mL. A high ($10^7$ yeasts/egg) or low inoculum ($10^5$ yeasts/egg) was applied onto the chorio-allantoic membrane at developmental day 10 via an artificial air chamber. In each experiment, the viability of 20 eggs per group (*Candida* or PBS control) was evaluated for 7 days by daily candling. Experiments were performed in duplicate. Surviving embryos were humanely terminated by chilling on ice at the end of the experiment. All experiments were performed in compliance with the German animal protection law. According to this, no specific approval was needed for work performed in avian embryos. All experiments were terminated on developmental day 18 at the latest.

## RESULTS

### Identification of a novel factor promoting species-specific chlamydospore formation

To uncover genes that contribute to the obvious differences in the virulence of *Candida albicans* and *Candida dubliniensis*, we screened a library of *C. albicans* genomic fragments integrated into a *C. dubliniensis* recipient strain. We especially screened for novel morphology-associated *C. albicans* genes, based on a finding that SLAD agar induces extensive chlamydospore production only in *C. dubliniensis* after prolonged incubation at 27–30°C [Fig. 1A and B, and (64)]. This was confirmed with chlamydospore-specific GFP reporter strains (34) (Fig. S1).

In total, about 20,000 transformants were tested under these conditions. We found smooth colonies of *C. dubliniensis* transformants that phenocopied the *C. albicans* colony appearance. One clone, CdINT4, showed a strong and reproducible reduction in sporulation and filamentation (Fig. 1C). Sequence analysis of the integrated *C. albicans* DNA fragment revealed an assembly of three adjacent open reading frames (orf19.2113, orf19.2114, and orf19.2115). To determine which orf was responsible for the phenotype, each (including up- and downstream regions) was separately integrated into *C. dubliniensis*. Only orf19.2115 (strain Cd2115) changed the colony morphology to mimic the original integrant strain on SLAD agar (Fig. 1D). The *S. cerevisiae* ortholog of orf19.2115, *TCD2*, encodes a tRNA-threonylcarbamoyladenosine dehydratase (13), which like many genes in *S. cerevisiae* has a paralog, *TCD1*. No paralogs of orf19.2115 seem to exist in *C. albicans* or *C. dubliniensis*. As the gene name *TCD2* is already in use in *C. albicans*, and considering the results described in the next sections, we named this gene *HMA1* (for **H**yper**m**odification of **A**denosine) and its *C. dubliniensis* ortholog (*CD36_15150*) Cd*HMA1* in this text. These two genes are syntenic, and their proteins are nearly identical, with 95.6% identities and 97.7% similarities in a BLASTP search (alignment in Fig. S2). We then created the deletion mutants *hma1Δ/Δ* and Cd*hma1Δ/Δ* and the revertants *hma1Δ/HMA1* (*C. albicans*) and Cd*hma1Δ/CdHMA1* (*C. dubliniensis*) by integration of one allele into the native locus. Finally, Cd*HMA1* was deleted in Cd2115 to obtain a *C. dubliniensis* Cd*HMA1*-deficient strain which carries only the *C. albicans HMA1* in one copy (Cd2115_Cd*hma1Δ/Δ*). All strains were verified by Southern blotting (Fig. S3).

### Hma1 is a regulator of morphogenesis in *C. albicans* and interacts with the TOR pathway

We had identified *HMA1* on a nitrogen-poor medium. The TOR pathway plays a central role in nutritional stress in eukaryotes (65). Hence, we tested the growth of the *C. albicans* wild-type SC5314, *hma1Δ/Δ*, and *hma1Δ/HMA1* in the presence of sublethal concentrations of the TOR antagonists rapamycin and caffeine. In YPD without additives, all strains grew fast and reached a plateau within 16 h (Fig. 2A). Addition of 7.5 nM rapamycin or 8 mM caffeine (Fig. 2B) increased the generation times in all strains, but the *C. albicans hma1Δ/Δ* strains were significantly less affected than the wild type. The complemented strain bearing one *HMA1* allele showed an intermediate growth.

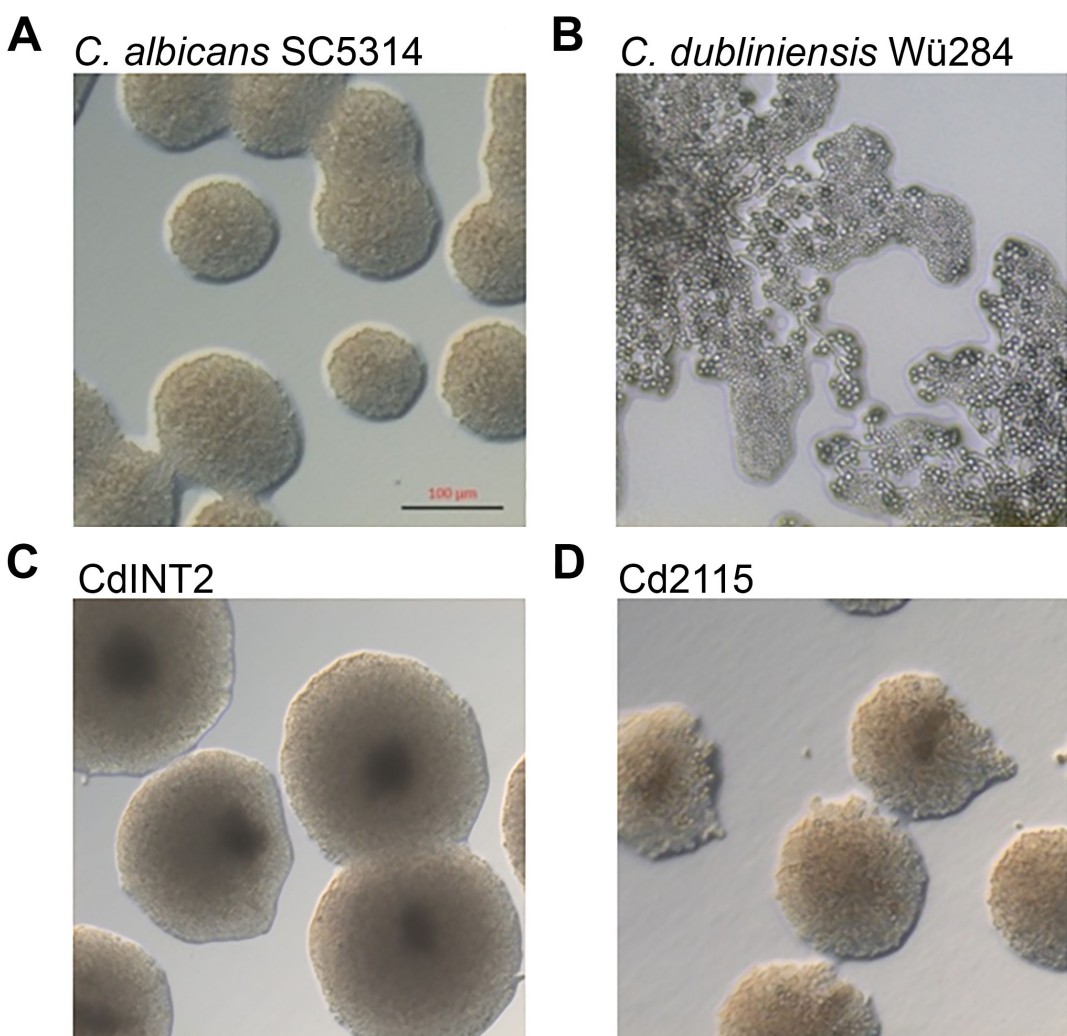

**A** *C. albicans* SC5314

**B** *C. dubliniensis* Wü284

**C** CdINT2

**D** Cd2115

**FIG 1** Screening of a *C. albicans* genomic library on SLAD agar. Cells were cultivated in liquid SD medium overnight (30°C, 180 rpm) and spread onto SLAD medium. Agar plates were incubated for 2 days at 30°C in the dark. (A) The *C. albicans* wild-type strain SC5314 grew in smooth colonies comprised yeasts, whereas (B) *C. dubliniensis* Wü284 formed pseudohyphae with end-terminal chlamydospores, identifiable by the rough colony morphology. (C) By screening the integrative genomic *C. albicans* library in *C. dubliniensis*, CdINT4 was isolated as a transformant, which resembled the smooth colony morphology of *C. albicans*. (D) Integration of *C. albicans* orf19.2115 (*HMA1*) into *C. dubliniensis* resulted in the strain Cd2115 that phenocopied the *C. albicans*-specific colony morphology. The scale bar represents 100 μm.

All tested *C. dubliniensis* strains grew equally well in YPD media (Fig. 2A), but in direct contrast to *C. albicans hma1*Δ/Δ, *C. dubliniensis Cdhma1*Δ/Δ strains exhibited an increased sensitivity to rapamycin and caffeine (Fig. 2B). Together, these findings indicate that Hma1 function interacts with Tor1-mediated signaling.

The TOR pathway is crucial for responding to the cellular nutritional status. Hence, we tested growth on the non-fermentable carbon sources glycerol and ethanol but found no effect of the *HMA1* deletion in either *Candida* species (Fig. S4A). Similarly, *C. albicans hma1*Δ/Δ showed no phenotypic effect when tested under oxidative (menadione), antifungal (oligomycin, antimycin A), cell wall (calcofluor white), and osmotic (NaCl) stress (Fig. S4B). This hints toward a specific role for *HMA1* in TOR-related morphology and nutrient signaling, rather than a global effect, similar to the phenotypes of $t^6$A-deficient *S. cerevisiae* (9).

Correct tRNA modifications optimize translational accuracy and consequently growth. Addition of the translation inhibitor cycloheximide in sublethal levels resulted in a

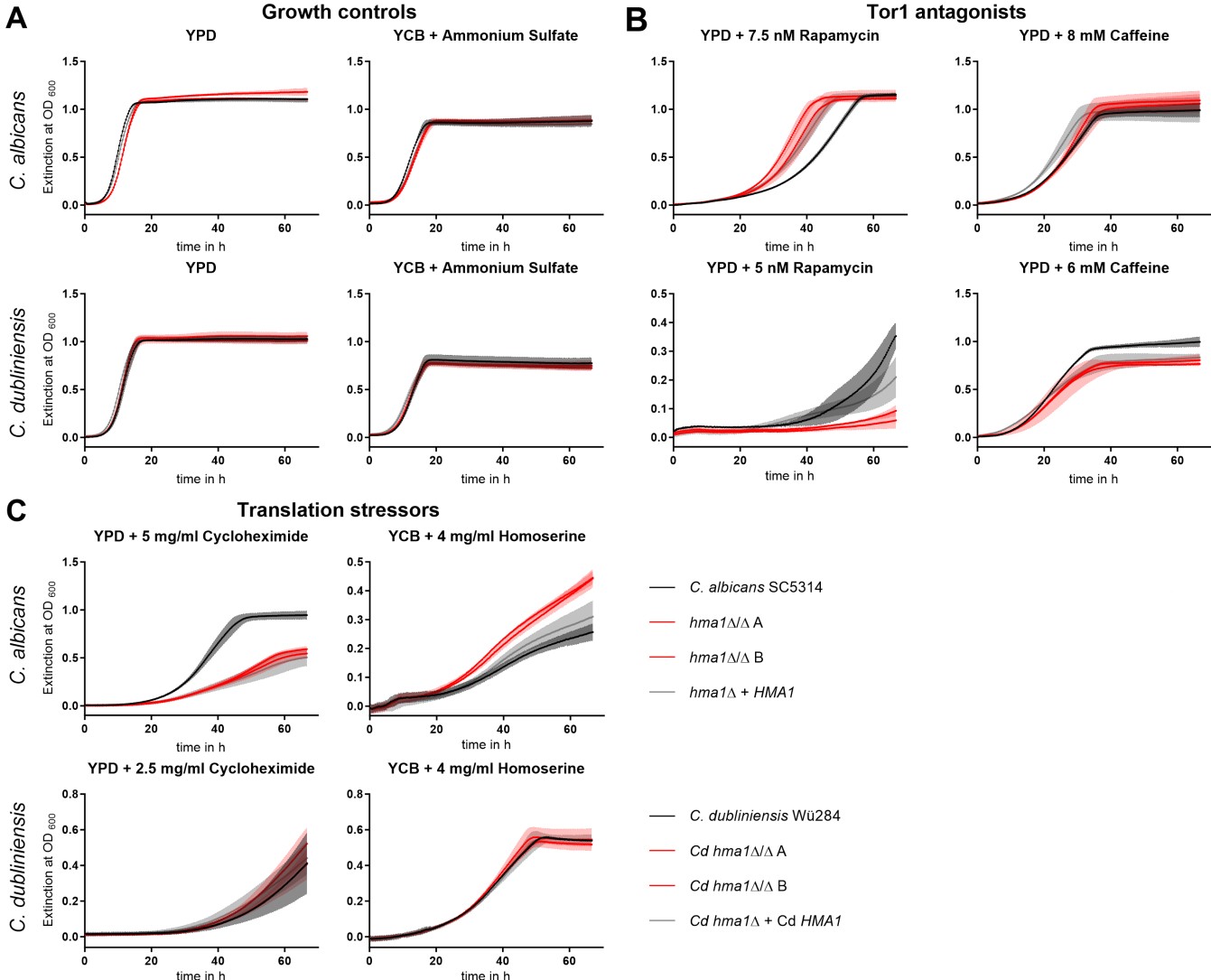

**FIG 2** Stress resistances test of *C. albicans* hma1Δ/Δ and *C. dubliniensis* Cd*hma1*Δ/Δ strains. *Candida* strains (*C. albicans*: SC5314, *hma1*Δ/Δ, *hma1*Δ/*HMA1* and *C. dubliniensis*: Wü284, Cd*hma1*Δ/Δ, Cd*hma1*Δ/*HMA1*) were grown overnight in YPD medium at 30°C. Growth of *C. albicans* strains was monitored over 65 h in (A) YPD and YCB plus 5 mg/mL ammonium sulfate, (B) YPD plus 7.5 nM rapamycin, YPD plus 8 mM caffeine, (C) YPD plus 5 mg/mL cycloheximide, YCB plus 4 mg/mL L-homoserine. Stressor concentrations were adapted for growth of *C. dubliniensis* strains to YPD plus 5 nM rapamycin, 6 mM caffeine, or 2.5 mg/mL cycloheximide. The temperature was set at 30°C for all conditions. Graphs show the mean ± SD of three independent biological replicates and two independently created *hma1*Δ/Δ strains for both species.

strong growth reduction of *hma1*Δ/Δ (Fig. 2C), suggesting an important role for Hma1 in translation. *C. dubliniensis* was generally more susceptible to cycloheximide, and no differences were evident between the effect on the wild type and the Cd*hma1*Δ/Δ mutants (Fig. 2C).

The threonine analog L-homoserine suppresses the growth defect of $t^6A$ tRNA modification-defective *S. cerevisiae* mutants (9). Similarly, *hma1*Δ/Δ grew better than the wild type and the complemented strain when 4 mg/mL L-homoserine was the sole nitrogen source, while the deletion of Cd*HMA1* in *C. dubliniensis* wild type had no effect (Fig. 2C). The control condition with ammonium sulfate supplementation revealed similar growth of mutants and parental strains (Fig. 2A). Metabolism of homoserine hyper-activates the protein degradation response (66), and our data therefore hint toward a possible link of *C. albicans HMA1* to this pathway.

Many tRNA modifications are mediators of heat stress responses (67, 68). In-line with these findings, a slight decrease in growth was observed for the mutants in both species at elevated temperatures (Fig. S5A and B). Therefore, we assume a species-independent role for *HMA1* during heat stress.

### *C. albicans HMA1* encodes a tRNA-modifying enzyme

Tcd2, a tRNA threonylcarbamoyladenosine dehydratase in *S. cerevisiae*, shows high similarity with *C. albicans* Hma1 (identities 60.2%, similarities 77.0%) and *C. dubliniensis* CdHma1 (61.4%, 78.4%). The overall structure of Tcd2 and Hma1 is also similar (Fig. 3A), suggesting similar functions. This enzyme targets $t^6A$ residues at position 37 ($t^6A_{37}$) specifically in tRNAs bearing an NNU anticodon, where it catalyzes the formation of an oxazolone ring ($ct^6A_{37}$) using ATP (Fig. 3B) (13).

To uncover the molecular function of Hma1, we cultivated *C. albicans* wild-type, $hma1\Delta/\Delta$, and revertant strains, as well as *C. dubliniensis* wild type in YPD at 30°C and 37°C and isolated tRNA from these strains. Subsequently, we analyzed the presence of $t^6A$ derivates by RNA mass spectrometry (48). Since $ct^6A$ is very labile (13), we determined the levels of its educt, $t^6A$ relative to a $^{15}N$-labeled spike-in (Fig. 3C). We found a striking increase (>50%) of $t^6A$ in both independent $hma1\Delta/\Delta$ strains. However, we did not observe a difference among the growth conditions or between the wild types of the two species. Importantly, $t^6A$ levels of the revertants were similar to the wild-type strains under both conditions. The total amount of $ct^6A$ was insufficient to obtain quantitative results due to its limited stability. However, we detected a weak MS1 signal indicating the presence of $ct^6A$ in 75 of 77 samples [49/50 (98%) in *C. albicans* and 27/27 (100%) in *C.*

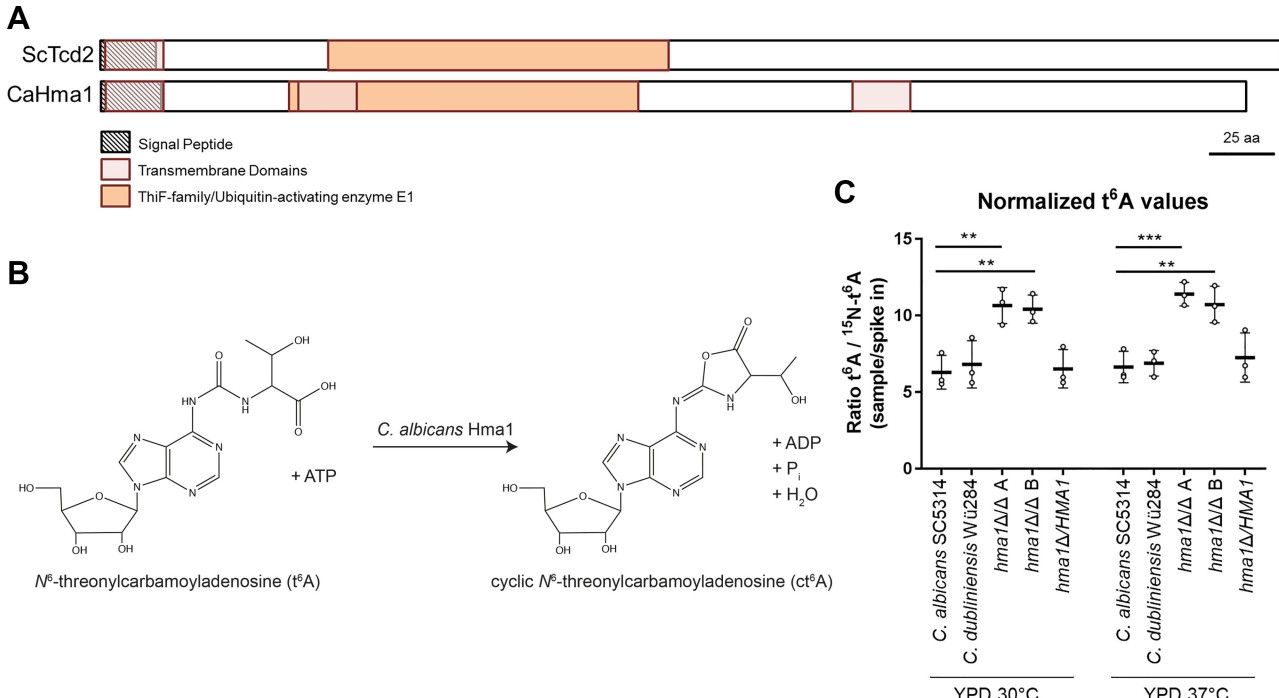

**FIG 3** Quantification of $t^6A_{37}$ modification in *Candida* species. (A) Structure of the proteins Tcd2 and Hma in *S. cerevisiae* and *C. albicans*, respectively. Important regions are highlighted to illustrate the overall structure. (B) Proposed model of cyclic $N^6$-threonylcarbamoyladenosine formation by Hma1 reaction in *C. albicans* (designed with ISIS/Draw2.1.4). (C) Overnight cultures of *Candida* strains (*C. albicans*: SC5314, $hma1\Delta/\Delta$ A/B, $hma1\Delta/HMA1$, and *C. dubliniensis* Wü284) were inoculated in fresh YPD medium ($OD_{600}$ of 0.2) and incubated for 4 h at 30°C or 37°C, cells were harvested, and RNA was isolated. Quantitative liquid chromatography mass spectrometry analysis of tRNA isolated from *Candida* strains was performed, and results of biological triplicates with their means were plotted relative to the spike in. The levels of $t^6A$ revealed a significant increase in the $hma1\Delta/\Delta$ A and B mutants compared to the *C. albicans* wild-type SC5314 in both test conditions. $t^6A$ levels of *C. dubliniensis* Wü284 and *C. albicans* $hma1\Delta/HMA1$ revertant remained at *C. albicans* wild-type level. Experiments were performed in biological triplicates, and statistical analyses used *t*-test with Šidák corrections for multiple comparisons (**$P \le 0.01$ and ***$P \le 0.001$).

*dubliniensis*, over all biological and technical replicates and conditions; Table S2]) from the wild types and the revertant but only in 2 of 50 (4%) samples taken from the deletion mutants. This signal was observed at the same retention time as a chemically synthesized nucleoside standard for ct$^6$A. Finally, the total levels of adenosine and other nucleosides remained essentially unchanged in all tRNA samples (Fig. S6). Altogether, these findings provide evidence for a lack of conversion of t$^6$A$_{37}$ to ct$^6$A$_{37}$ upon deletion of *HMA1* in *C. albicans*. This is in-line with findings in *S. cerevisiae* and suggests that Hma1 enables the conversion of t$^6$A to ct$^6$A (Fig. 3B).

## Hma1 plays a species-specific role during hyphae formation

Given the role in morphology we have established for *HMA1*, we next assayed the filamentation abilities of all strains with minimal or nutrient-deprived (water and SLAD) agar and complex media (YPD, Spider, and chocolate agar, and water + 10% fetal calf serum liquid medium).

The *C. albicans hma1Δ/Δ* deletion mutants showed clearly impaired hyphae fringes at the colony borders on chocolate, Spider, and YPD agar (Fig. 4A through C). The phenotype was nearly fully restored in the *hma1Δ/HMA1* revertant. In contrast, in all strains, the central wrinkling of the colonies was comparable to the wild type (Fig. 4A through C). This indicates a specific reduction in longer hyphal elements, which invade the agar, rather than a general loss of filamentation. Similarly, using an XTT assay, we found no difference in the biofilm formation by the *C. albicans hma1Δ/Δ* mutants (data not shown). During formation of biofilms, hyphae growth is important, but no invasion takes place.

In general, *C. dubliniensis* shows little to no hyphae formation on standard agar plates. Only on minimal medium agar at 37°C, a late robust hyphae formation without central wrinkling was observed (Fig. S7A through C). However, the deletion, revertant, and transformant mutants exhibited no difference to the *C. dubliniensis* wild type, indicating that CdHma1 is dispensable in *C. dubliniensis* filamentation on agar plates.

We also tested both species for filamentation in liquid medium using water with 10% FCS. Hyphae length of *C. albicans* wild type reached on average 60.5 µM after 4 h, whereas *hma1Δ/Δ* hyphae were significantly shorter (40.2 and 39.6 µm for the two independent mutants; Fig. 5A). The complemented strain reached almost wild-type levels (54.8 µm). In stark contrast, the hyphae length of Cd*hma1Δ/Δ* in water-serum medium increased significantly (129% and 131%), which was largely reversed by

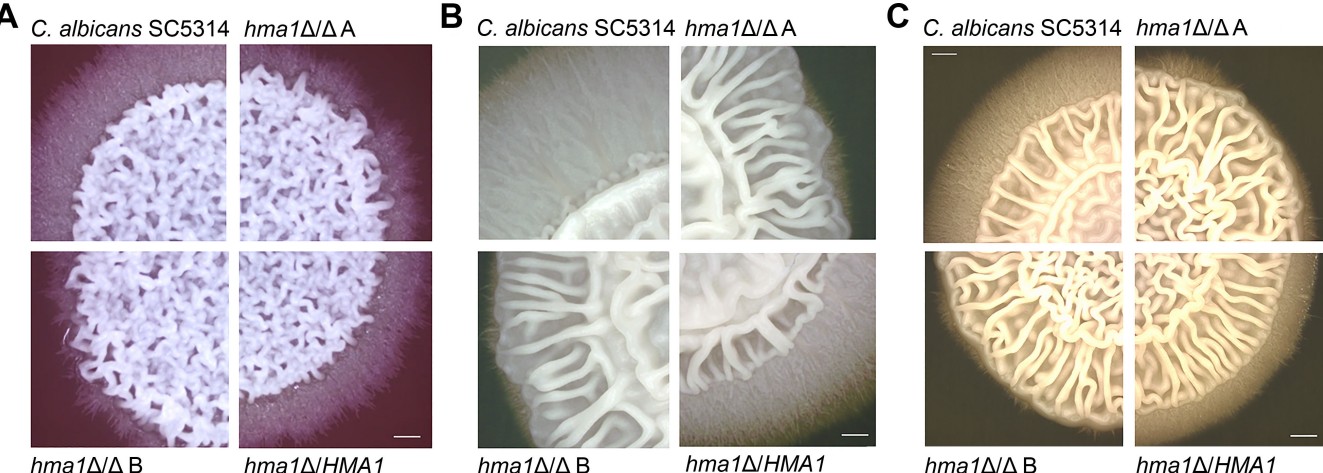

**FIG 4** Filamentation of *C. albicans* strains on solid agar. *C. albicans* cells from a YPD overnight culture were washed, and 10$^5$ cells of each strain were spotted on (A) chocolate agar, (B) Spider agar, or (C) YPD agar. Plates were incubated at 37°C for 5 days. Each medium induced wrinkling in the center of the colonies and extensive hyphae formation by the *C. albicans* wild-type SC5314. The *hma1Δ/Δ* mutant strains were severely impaired in formation of the hyphal fringe, but central wrinkles remained. The scale bar represents 0.5 cm.

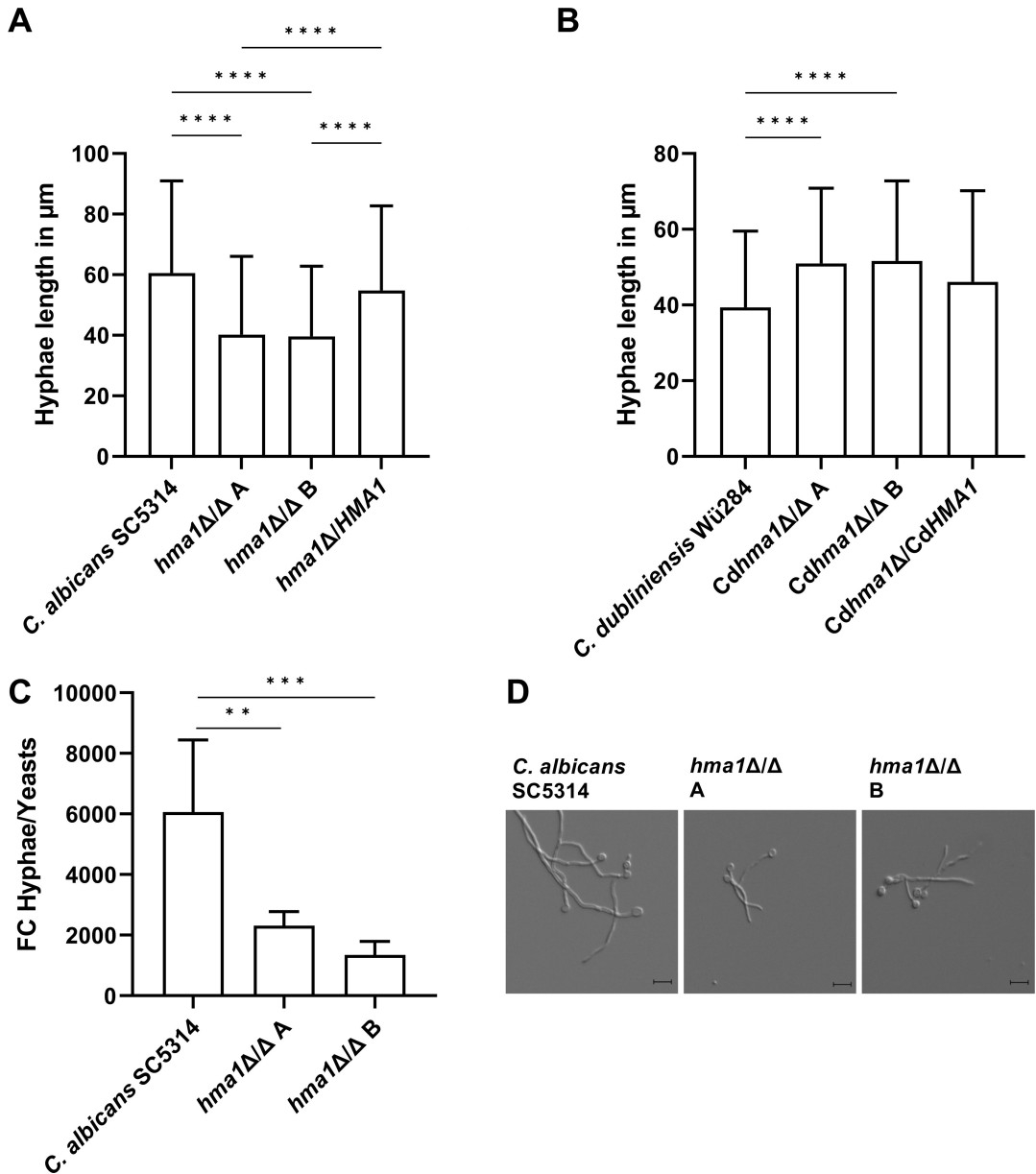

**FIG 5** Length of hyphae of *C. albicans* and *C. dubliniensis* strains incubated in water plus 10% serum. In total, $10^4$ *Candida* cells were incubated in water plus 10% FCS for 4 h at 37°C, and atmospheric $CO_2$ content and hyphal length were measured microscopically. (A) *C. albicans* hma1Δ/Δ mutants had significantly shorter hyphae than the wild-type strain. (B) *C. dubliniensis* Cd*hma1*Δ/Δ mutants had longer filaments compared to the wild type. The column bars show mean values with standard deviation. (C) Transcriptional upregulation of the toxin-coding *ECE1* gene upon hyphae formation. The transcript levels are significantly lower in the mutant, in-line with (D) the reduced hyphae length of the mutants as seen in micrographs. Statistical analysis used one-way ANOVA test with Tukey's multiple comparisons test (****$P \leq 0.0001$). Sample size (N) is 3 biological replicates (A–C) with at least 50 individual measurements (A, B).

re-introducing Cd*HMA1* (Fig. 5B). This shows that the effect of Hma1 differs strikingly between the two species and that *C. albicans* Hma1 promotes growth of long hyphae, whereas *C. dubliniensis* Hma1 suppresses it. When we investigated the transcription of *ECE1*, the gene coding for the *C. albicans* toxin candidalysin (69), we found a strong reduction in its upregulation upon hyphae formation (Fig. 5C). As *ECE1* transcript levels are strongly correlated with hyphal length (70, 71), these data support our morphological findings (Fig. 5D). Since hyphae formation and the secretion of *ECE1*-encoded candidalysin are central to *C. albicans* pathogenesis, we continued to investigate Hma1's role in virulence.

## *HMA1* deletion reduces adherence and invasiveness into oral epithelial cells

To investigate adhesion and invasion as the two primary steps in *Candida* infections, we infected the human oral epithelial TR146 cell line with *C. albicans* wild type, *hma1Δ/Δ*, and revertant strains. After 3 h, 25.7% of the wild-type cells, but only 13.0% and 15.4% of the two independent *hma1Δ/Δ* strains were found adhered to host cells, a reduction by roughly half. This phenotype was mostly reversed in *hma1Δ/HMA1,* at 21.2% adherence (Fig. 6A).

Furthermore, we determined invasiveness, defined here as the proportion of hyphae with an invasive part relative to all adherent fungal cells. , 36.0% of the *C. albicans* wild-type cells were invasive, while the *hma1Δ/Δ* mutants were significantly reduced (24.0% and 26.5%; Fig. 6B). The complemented strain regained wild-type levels of invasiveness at 41.3%. Similarly, the total hyphal length in the *hma1Δ/Δ* strains was reduced to 76.2% and 80.6% of wild type (Fig. 6C), in agreement with our *in vitro* water-serum medium data. Interestingly, if a hypha had successfully invaded epithelial cells, the length of the invasive part did not differ significantly (Fig. 6D). This indicates that adhered growth and initial invasion, but not subsequent elongation, inside the host cell were inhibited.

## Lack of Hma1 attenuates virulence of *C. albicans in vivo*

Given the *in vitro* effects of the *HMA1* deletion on hyphae, we predicted an impact of the loss of Hma1 (and hence tRNA modifications) on *C. albicans* virulence. To test this hypothesis, we employed an embryonated chicken egg infection model (32), where we infected eggs with high or low doses of *C. albicans* wild type, *hma1Δ/Δ*, or *hma1Δ/HMA1*. Vitality of the avian embryos acted as the read-out for fungal virulence over 7 days.

As expected, infection with high doses of *C. albicans* resulted in high lethality within 3 days with all strains (Fig. S8A). However, at a lower infection dose where fungal growth became more relevant, significant differences were observed. Infection resulted in a mortality of 92.5% after day 7 (Fig. 7) for the wild type, but only 27.5% for one *hma1Δ/Δ* strain, whose isogenic *hma1Δ/HMA1* heterozygous strain reverted to an intermediate 52.5%. A second, independent *hma1Δ/Δ* mutant (strain A) was similarly severely attenuated in virulence, albeit at a slightly lower overall level (Fig. 7). Together, these data show that the Hma1 is required for full virulence of *C. albicans*.

## Ribosome profiling reveals translational stalling on ANN and non-ANN codons in a *hma1Δ/Δ* mutant

We found that the loss of Hma1 affects $ct^6A_{37}$ tRNA modification formation in *C. albicans*. Therefore, we performed ribosome profiling to assess mRNA translation in wild type and *hma1Δ/Δ*. When cultivated at 30°C—a temperature non-conducive to hyphae formation—we did not observe any difference in codon occupancy patterns for both strains (Fig. 8A). However, incubation at 37°C led to an increase of ribosome occupancy of several codons for the *hma1Δ/Δ* mutant. Half of the 16 codons with predicted $t^6A$ modifications differed significantly between wild-type and *hma1Δ/Δ* mutant, which supports the idea that Hma1 acts in the formation of $ct^6A_{37}$ (Fig. 8B). Surprisingly, a large subset of non-ANN codons was also affected. In particular codons which code for arginine or histidine exhibited increased ribosomes occupancy in the *hma1Δ/Δ* strain, as did non-canonical start codons (CTG, GTG, TTG). This may indicate an increased number of non-AUG translation initiation events.

Finally, we analyzed differences in the overall translatome between *hma1Δ/Δ* and wild type. At 37°C, we found 731 transcripts to be differentially translated between wild type and *hma1Δ/Δ*. Of these, 182 transcripts showed more ribosome reads in the wild type, with an enrichment of the GO processes "filamentation," "aggregation," and "biofilm formation" (Fig. 8C). These processes are therefore likely facilitated by Hma1 function. Among the more abundantly found transcripts are those of the adhesin gene *ALS1* and the G1 cyclin-related gene *HGC1*, which is involved in hyphal morphogenesis.

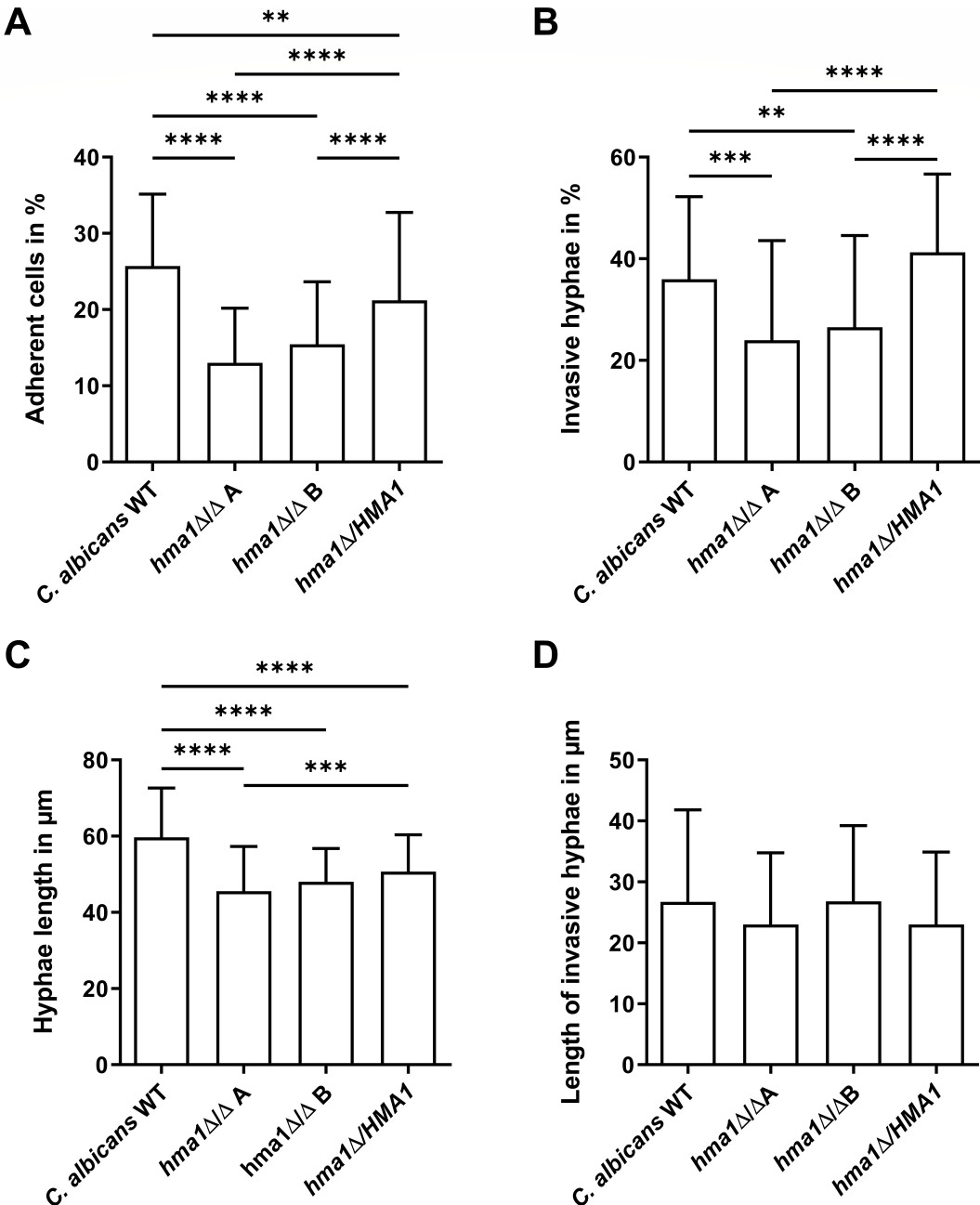

**FIG 6** *C. albicans* *hma1*Δ/Δ mutants are defective in adhesion and invasion of oral epithelial cells. *C. albicans* cells were washed, and an MOI of 0.4 was used to infect 2-day-old TR146 cells. After co-incubation of fungal and human cells for 3 h, non-adherent cells were rinsed off, and samples were fixed for staining and microscopic analysis. *C. albicans* *hma1*Δ/Δ mutants were (A) less adherent and (B) less invasive. (C) The total hyphae length was reduced for *hma1*Δ/Δ strains but (D) unaltered for the invasive parts of hyphae. The column bars show mean values with standard deviation. Statistical analysis used one-way ANOVA test with Tukey's multiple comparisons test (**$P \leq 0.01$, ***$P \leq 0.001$, and ****$P \leq 0.0001$). Sample size (N) corresponds to 3 biological replicates with 36 individual measurements of image sections for A and 24 for B and 50 individual measurements of hyphae lengths for C and D. (For D, hyphae without an invasive part were not included.)

Transcripts that were less abundant in the wild-type translatome were enriched for metabolic categories, especially lipid oxidation, glyoxylate cycle, and respiration (Fig. 8D).

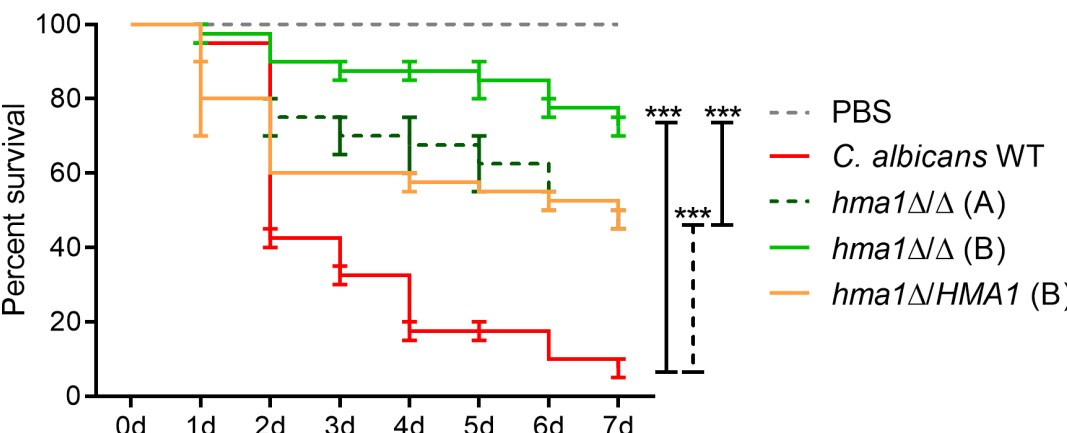

**FIG 7** Virulence of the *C. albicans* wild-type strain SC5314, *hma1Δ/Δ* mutants, and the *hma1Δ/HMA1* revertant in infected chicken *embryos*. Survival after infection is depicted as Kaplan-Meyer plots. There were 20 chicken embryos per group per experiment, and the combined results of two independent experiments are shown. The *hma1Δ/Δ* mutants exhibited significantly attenuated virulence (***$P < 0.001$) compared with the wild type. The reconstituted mutant *hma1Δ/HMA1* did not fully revert the virulent phenotype but led to a significantly (***$P < 0.001$) reduced survival compared to its parent, *hma1Δ/Δ* (B). All significances were calculated by log-rank (Mantel-Cox) test.

## DISCUSSION

### Cross-species comparison is a powerful tool to elucidate virulence factors

We started our investigation by exploring the genetic differences between *C. albicans* and its closely related sister species, *C. dubliniensis*. During evolution, (sub)populations of microorganisms adapt to novel niches, and new, beneficial genetic attributes can get fixed in the population (72), such as the acquisition of specific pathogenic features in response to an adaptation to a new host. Cross-species comparisons therefore can be a powerful tool to obtain novel insights into the genetic basis of species-specific lifestyles of related microorganisms.

While *C. albicans* appears to have acquired a number of specific virulence factors like hydrolytic enzymes or hyphae-associated genes (27), the less pathogenic *C. dubliniensis* underwent reductive evolution (26). Similarly, a cross-species transcriptome comparison of these *Candida* species has revealed differences in expression of orthologous, virulence-associated genes like *ECE1* (69, 73). In our cross-species morphological screen (37), we verified SLAD agar as an improved alternative to the often ill-defined classical chlamydospore induction media (28, 74). Its consistent and reproducible composition makes this synthetic medium an excellent candidate for discriminative *Candida* diagnostics.

The approach to integrate a genomic *C. albicans* library into *C. dubliniensis* was first used by Staib and Morschhäuser to discover the role of Nrg1 as a major *C. albicans* chlamydospore repressor (37). In fact, we found a similar phenotype by integration of the previously uncharacterized *C. albicans* gene *HMA1*. The resulting strain was impaired in chlamydospore and pseudohyphae formation. However, Hma1 is not a transcription factor but instead showed high similarities with the *S. cerevisiae* tRNA threonylcarbamoyladenosine dehydratase Tcd2. Interestingly, a previous large-scale screen in *S. cerevisiae* revealed that overexpression of *TCD2* increased invasive and pseudohyphal growth on nitrogen-sufficient medium (75), a phenotype that is to that of *C. albicans hma1Δ/Δ* mutants.

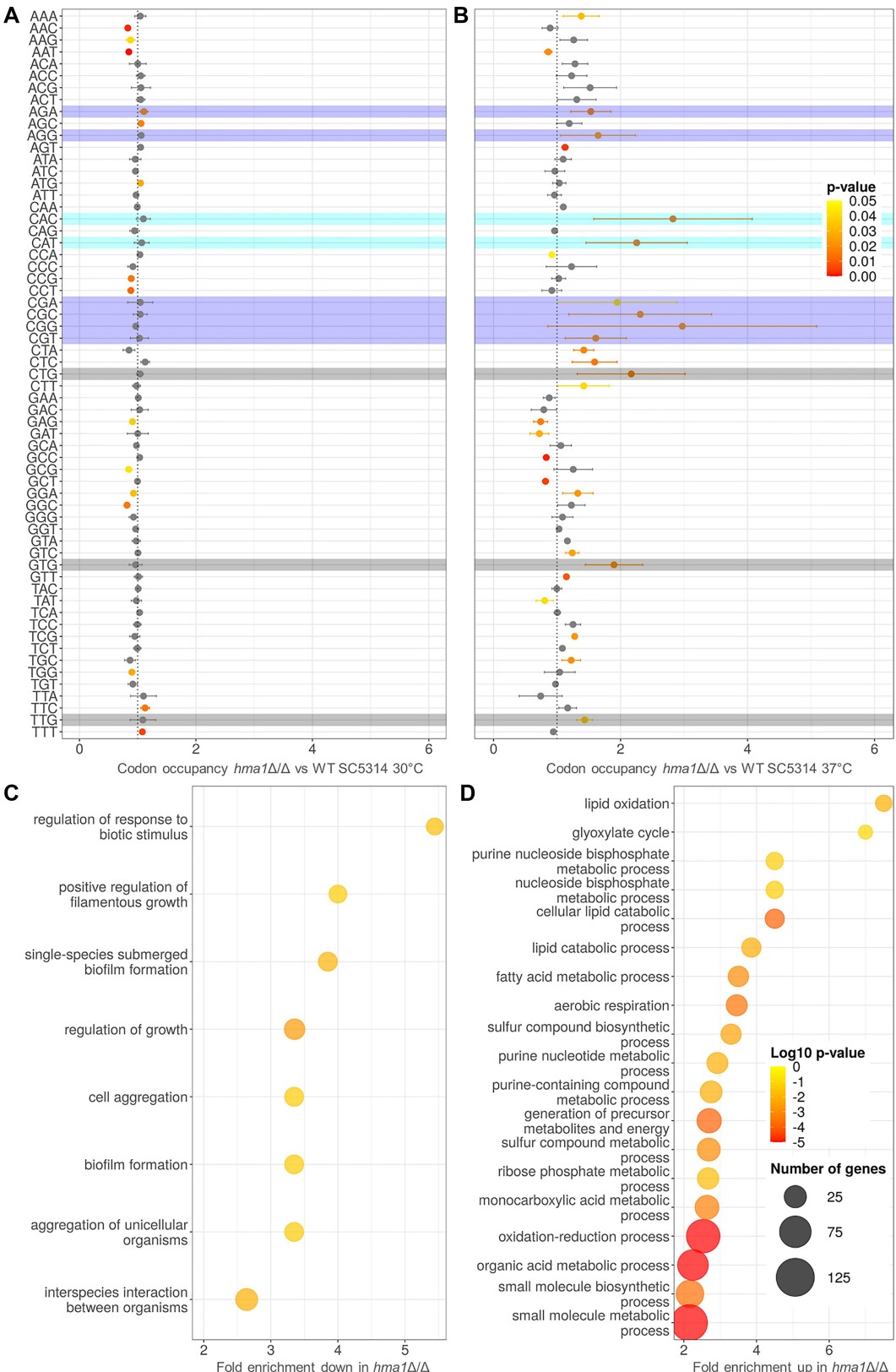

**FIG 8** Ribosome profiling of *C. albicans* hma1Δ/Δ mutant vs wild-type SC5314. *C. albicans* strains were incubated in YPD medium at 30°C and 37°C, and samples were processed for ribosomal profiling to analyze the codon-specific ribosome stalling and for ribosome sequencing. (A, B) Codon-specific changes in ribosome occupancy in *C. albicans* hma1Δ/Δ in comparison to wild-type SC5314 (geometric mean ± GSD; *n* = 3). Symbol color reflects statistically significant changes in

**FIG 8** (Continued)

codon occupancy (two–tailed student's $t$-test, $P_{adj}$ <0.05). Selected tRNA isoacceptors were labeled for arginine charged in violet, histidine charged in turquoise, and gray highlights non-canonical start codons. (C, D) Ribosome-protected fragments of the *C. albicans hma1Δ/Δ* mutant and the wild-type SC5314 (both cultivated at 37°C) were sequenced to determine differentially abundant mRNA transcripts. For these, enrichment of gene ontology terms was determined (log$_2$FC ± 1, $P_{adj}$ <0.05) and summarized using the REVIGO tool (63). Log$_{10}$ $P$-values, enrichment scores, and gene list sizes for enriched GO terms with less (C) or more (D) abundant transcripts in the *hma1Δ/Δ* vs the wild-type SC5314 are shown.

## Wide-ranging consequences of t⁶A modifications

The importance of tRNA modifications for cellular processes has been extensively studied in baker's yeast and humans, as many genetic diseases, including cancer, diabetes, and several neuropathies, are caused by defective tRNA modifications (76, 77), yet the effector mechanisms are poorly understood. Miyauchi et al. discovered the cyclic t⁶A$_{37}$ and showed its unexpected wide prevalence in bacteria, fungi, and plants (13). Further investigations into this condensed compound are lacking, but the structural and steric similarities to the t⁶A precursor molecule allow for functional assumptions. The t⁶A modification is one of the few conserved tRNA modifications which occurs in all domains of life and is found in all eukaryotic initiator tRNAs (78). Defects in the t⁶A synthesis pathway are lethal in many prokaryotes (79), including the pathogens *Mycoplasma pulmonis* (80), *Salmonella* Typhi (81), and *Staphylococcus aureus* (82). In fact, strategies to target t⁶A biosynthesis for antibiotic therapy have been considered (10). Similarly, t⁶A-defective mutants in *S. cerevisiae* are retarded in growth (9), while ct⁶A$_{37}$-defective strains grow normally (13, 83). This was also reflected by the robust growth that we observed in *hma1Δ/Δ* mutants in *C. albicans* and *C. dubliniensis*. However, treatment with the translation inhibitor cycloheximide reduced the growth of *C. albicans hma1Δ/Δ* mutants, indicating its function in correct translation, which was not observed in *C. dubliniensis*. Species-specific responses to translational perturbations have been described previously in yeasts and may contribute to this observation (59).

The molecular function of Hma1 as a tRNA threonylcarbamoyladenosine dehydratase of *C. albicans* was demonstrated by the accumulation of the ct⁶A precursor, t⁶A in the deletion mutants. Direct detection of ct⁶A, the final product of the proposed Hma1 reaction, could not be achieved quantitatively due to its low stability and its inefficient ionization during mass spectrometric measurements (13). However, we were able to detect trace amounts in the wild type and the revertant but not in the deletion mutant. Based on these findings, we conclude that Hma1 catalyzes the formation of ct⁶A from t⁶A in *C. albicans*—similar to Tcd2, its *S. cerevisiae* homolog—and is thus critical for correct tRNA modifications in this fungus. The ct⁶A modification is thought to facilitate proper translation of ANN codon stretches in *S. cerevisiae*. Surprisingly, numerous codons were slowed down in translation including ANN and non-ANN codons. The highly affected tRNA isoacceptors were those charged with arginine and histidine, which could be a secondary effect due to a general diminished translational activity. The enrichment of non-canonical start codons like CTG could reflect an alternative translational scanning mechanism in response to cellular stress (84). Alternatively, unmodified initiator tRNA may interfere with the decoding of near-cognate start codons.

In *S. cerevisiae*, Tcd2 is associated with the KEOPS complex, and the knockout of members of this complex blocks or diminishes t⁶A formation (83). Threonine, in addition to carbonate and ATP, is a substrate in the initial steps of t⁶A formation, and availability of threonine is therefore a determinant of t⁶A modification levels (10). Supplementation with L-homoserine, an isomer of threonine, consequently increased the growth rate of *S. cerevisiae tcd2Δ* (9). In *C. albicans*, *hma1Δ/Δ* mutants even outperformed the parental strain when exposed to L-homoserine. Key phenotypes of *S. cerevisiae tcd2Δ* mutants include elevated sensitivity to heat stress (37°C) (85) and the inability to use non-fermentable carbon sources (13, 86). As shown before, the connection between heat resistance and tRNA modification differs among yeast species depending on their evolutionary history and ecological niche (68). However, the effect was observable

in the *Candida hma1Δ/Δ* deletion mutants as a growth defect at elevated, fever-like temperatures. Moreover *C. albicans* codon occupancy was remarkably different in the *hma1Δ/Δ* mutant at 37°C, indicating an important role in normal protein biosynthesis of *C. albicans* at this temperature, with evident consequences for colonization and infections of endothermic hosts like humans.

Baker's yeast and Crabtree-negative *Candida* species differ in their primary carbon catabolism (87), and we consequently observed no correlation between ct$^6$A formation and respiratory growth for the latter. Nevertheless, ribosome profiling showed that genes implicated in lipid and purine metabolism were differentially translated in the *hma1Δ/Δ* mutant. Finally, levels of many tRNA modifications have been proposed as possible internal markers for exposure to cellular stressors (15, 88). However, we did not observe growth defects for *C. albicans hma1Δ/Δ* mutants in the presence of osmotic, oxidative, or cell wall stressors. This suggests that Hma1 is not a mediator of the global stress response.

## tRNA modification levels affect the TOR pathway and influence morphology

TOR signaling directs complex cellular networks in response to nutrient availability (89). Generally, *C. dubliniensis* was more sensitive to rapamycin in our growth assays. The reason is not clear, but we note that differences in TOR activation have been linked to the lower rate of filamentation in *C. dubliniensis* compared to *C. albicans* (27). Potentially, the basal TOR activity level or the downstream signaling—including modulation by Hma1—may differ between the two fungi (27), making *C. dubliniensis* more sensitive to perturbations.

The link between the TOR pathway and tRNA modifications like t$^6$A is conserved among the eukaryotes: A loss of modifications at the ASL decreases rapamycin resistance in *S. cerevisiae* (11, 19), KEOPS complex mutants in *Drosophila* resemble TOR pathway-deficient mutants (90), disturbance of tRNA-modifying enzymes reduces the TOR-driven cell growth in *Arabidopsis thaliana* (91), and a hypomodified adenosine residue at position 37 of tRNAs is considered a risk factor for type 2 diabetes in humans as a consequence of mTOR overactivation (92, 93). Interestingly, loss of Cd*HMA1* in *C. dubliniensis* elicited rapamycin sensitivity similar to *S. cerevisiae* t$^6$A-deficient strains (9). The opposite was found for the *hma1Δ/Δ* gene deletion in *C. albicans*, which even enhanced resistance to TOR-antagonistic substances. Hence, Hma1 acts—in interplay with the TOR-pathway—in a species-specific manner in these sister species. Whether this is due to intrinsic differences in the enzymatic activity between the two orthologs, different expression pattern or the underlying biological differences between the two species, is an interesting open question for future research.

In addition to the generally conserved TOR functions in nutrient signaling, reduced Tor1 activity contributes to *C. albicans* yeast-to-hyphae transition via nucleosome repositioning that allows expression of hyphae-associated genes (94). This was reflected by the role of Hma1 in polymorphism: A function in morphological transitions was initially indicated by the integration of *C. albicans HMA1* into *C. dubliniensis*, which changed its morphology under nitrogen and amino acid deficiency. Deletion of *HMA1* in *C. albicans* drastically decreased the conversion of t$^6$A into ct$^6$A, concomitant with a severe decrease in hyphae length and invasiveness into oral epithelial cells. Interestingly, this effect was speciesspecific like the rapamycin response, as *C. dubliniensis* Cd*hma1Δ/Δ* mutants formed unaltered hyphae on solid media and even longer filaments in the presence of serum. It cannot be excluded that in addition to the TOR pathway, other regulators of hyphae formation are affected by Hma1 directly or indirectly. Especially the TOR pathway can affect negatively and positively acting transcription factors of hyphae-associated genes, like Nrg1 or Efg1 (64). Future studies may help to disentangle the relative contributions of these regulators to the tRNA modification-driven pheno-type.

Filamentation responses of *C. albicans* are known to depend on the medium used and often differ significantly between liquid and solid media depending on their composition

(95). On (semi-) solid media, TOR pathway activity can increase the hyphae formation of *C. albicans* (64). In contrast, in liquid medium, Tor1 activity favors growth in yeast form of both *C. albicans* and *C. dubliniensis*, but hyphae elongation can be enhanced by supplementing rapamycin (29, 94, 96). How does Hma1 compare to these Tor1 effects? In our *C. albicans* experiments, Hma1 was required for the formation of a hyphal colony fringe specifically on nutrient-rich agar, where rapamycin is known to have no effect (64). Therefore, *C. albicans* Hma1 appears to be required under conditions where Tor1 activity is dispensable for hyphal growth. Based on these data, we suggest opposing roles for Tor1 and Hma1-mediated ct⁶A modification in *C. albicans*. Interestingly, in *C. dubliniensis* Cd*hma1*Δ/Δ mutants, both the effect on rapamycin resistance and on hyphae formation was reversed in comparison to *C. albicans HMA1* deletion. It further points toward an evolutionary difference in the mutual relationship and the relative contributions of the *HMA1* and TOR pathway activity in the two species. The generally higher propensity of *C. albicans* to form hyphae, compared to *C. dubliniensis*, can thus be attributed in part to the differences in Hma1 activity, as demonstrated by our gene deletion and transfer experiments. Previously published data show that there is no difference in transcript levels of the *HMA1* orthologs between *C. albicans* and *C. dubliniensis*, even under conditions where only *C. albicans* filaments (73). Together with our gene transfer experiments, this indicates that Hma1 function, rather than expression level, may be at the core of the species specificity. It cannot be excluded, of course, that expression levels vary under specific conditions like nutrient availability, with consequences for tRNA modification levels and hyphae formation.

The influence of translation efficiency on *C. albicans* filamentation is barely understood. Isolated data point to a possible link between tRNA modification and morphology in fungi: low modification levels at the tRNA wobble anticodon in *S. cerevisiae* elicit defects in starvation-induced agar invasion (97), and genes coding for tRNA-modifying enzymes (*UBA4*, *URM1*, *NCS2*, *NCS6*) were enriched in a systematic screening of *S. cerevisiae* deletion mutants with altered colony morphology on solid agar (98). The integrity of tRNA modifications can affect the development of morphological structures in other organisms, from filamentous morphologies in *Salmonella* (99) to alterations in the auxin-regulated morphogenesis in *Arabidopsis* (100). Our ribosome profiling in *C. albicans* identified numerous morphology-associated transcripts enriched in wild-type translation, including key regulators for filamentation and biofilm formation: Hgc1, Brg1, Ras1, and Tec1 (94, 101–103). Recently, a reduced translational efficiency of highly expressed hyphae-associated genes of *C. albicans* has been reported, indicating a fine-tuning mechanism for translation of these virulence-related genes (104). This process could be regulated in part by tRNA modification patterns, such as ct⁶A.

In fact, overexpression of the *HMA1* ortholog *TCD2* in *S. cerevisiae* is known to result in enhanced pseudohyphal growth as well as increased invasion into nitrogen-sufficient agar (75). Similarly, the reduced hyphae length of *C. albicans hma1*Δ/Δ was accompanied by impaired invasiveness on agar plates and, importantly, on human epithelium, a phenotype that is linked to general virulence potential (41).

## tRNA modifications are required for full virulence

In most bacteria, a defect in t⁶A modification is lethal (14), and altered levels of prenylation of i⁶A$_{37}$ were found it to be crucial to the fitness and virulence of extraintestinal pathogenic *Escherichia coli* (105). Modifications that target the tRNA wobble bases are known to be essential for the expression of virulence factors of gram-negative bacteria like *Pseudomonas aeruginosa* (106). In *Samonella* Typhimurium, disrupted tRNA modification reduces expression of flagellar and secretion type III genes (99). Little is known on the role of tRNA modifications for fungal pathogenesis, but the plant pathogen *Colletotrichum lagenarium* failed to penetrate host tissue if 7-methylguanosine formation was defective (107)—a phenotype analogous to our *C. albicans hma1*Δ/Δ mutants. Furthermore, the levels of wobble uridine tRNA modification were recently shown to modulate the virulence of infectious strains of *S. cerevisiae* and *C. albicans* (108).

Recently, *C. albicans* Hma1 appeared as a candidate in a large phenotypic screening for damage factors of intestinal epithelia cells (109). Our finding that *ECE1* transcript levels are reduced in the mutant, in-line with the shorter hyphae, offers an explanation for this effect. Lower *ECE1* transcription likely results in less secretion of candidalysin, the *C. albicans* toxin that is mainly responsible for epithelial cell damage (39, 69). Together with the reduced adhesion and invasion properties during infection of human oral epithelia in our experiments, this suggests a general virulence defect of these mutants. As such, this would add another layer to the intricate regulation of hyphae formation, which is considered to be a central virulence mechanism of *C. albicans*. In addition to the established transcriptional and post-translational regulation, we here saw a co-translational regulation—a layer that differentiates the less pathogenic *C. dubliniensis* from *C. albicans*.

We hence used an alternative infection model to test the virulence potential of our tRNA modification-defective mutants: embryonated chicken eggs, a model which is known to correlate well with mice infections and which is suitable to predict progress of infection in mammals (110). In this avian model infection with $ct^6A$-defective strains significantly reduced mortality rates, supporting our concept of a role for tRNA modifications for fully expressing *Candida* virulence factors. Importantly, our *in silico* analyses showed Hma1 to be fungal specific without a human ortholog. Therefore, tRNA-modifying enzymes could potentially serve as future antifungal targets.

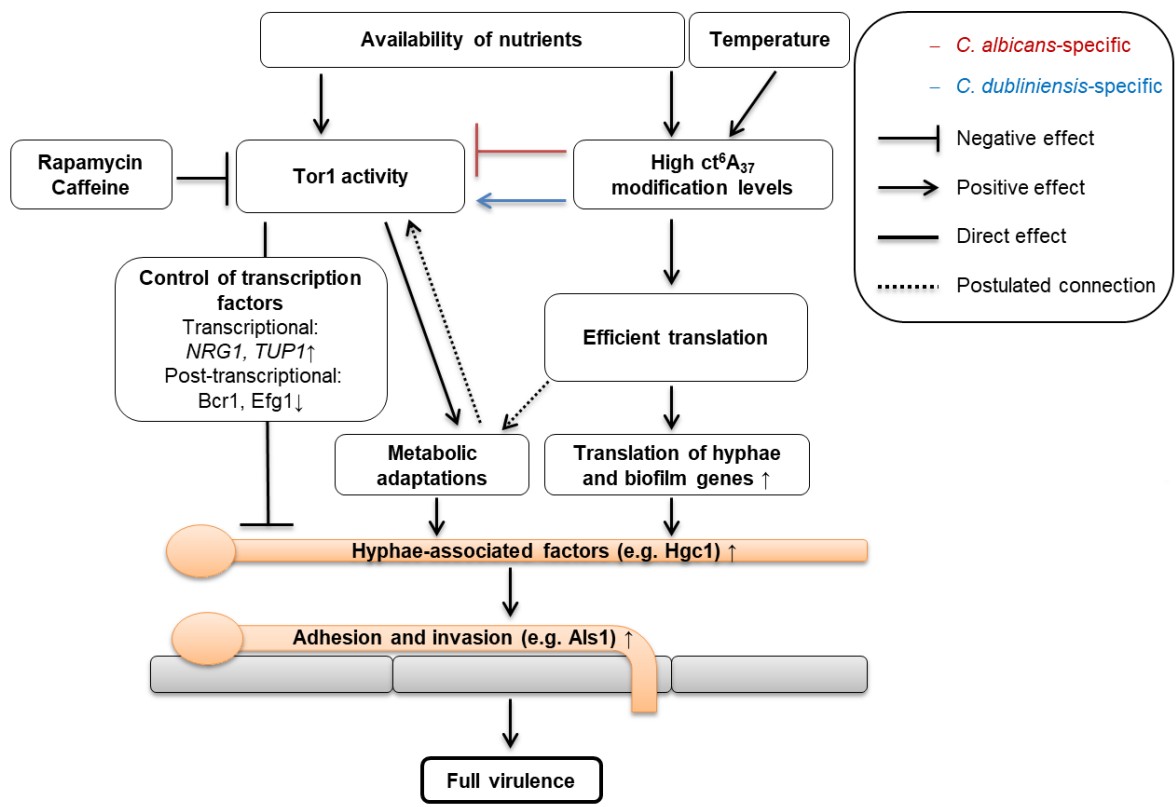

**FIG 9** Model presenting the impact of $ct^6A_{37}$ tRNA modification levels on pathogenicity-related features in *C. albicans* and *C. dubliniensis*. Environmental signals like high abundance of nutrient sources or a temperature of 37°C increase $ct^6A$ tRNA modification levels and Tor1 activity. The latter affects filamentation intensity of *C. albicans* and *C. dubliniensis* via transcriptional and post-translational regulation (64), but addition of rapamycin or caffeine inhibits this pathway. Hma1 interacts with the TOR pathway in a species-specific manner and impacts the translation of morphology-, metabolism-, and virulence-associated factors. The interplay of all pathways positively controls filamentation of *C. albicans* in $ct^6A$-replete cells and is necessary for proper adhesion to and invasion into surfaces like epithelia. Via these pathways, tRNA modifications strongly influence the biologically relevant outcome, virulence—as seen by the correlation between $ct^6A$ levels in *C. albicans* and chicken embryo mortality.

## Conclusion

tRNAs, with their modifications, link the transcriptome and the proteome of the cell. So far, tRNA modifications have not been implicated in the virulence and morphogenesis of human fungal pathogens. Based on our data, we propose a model (Fig. 9) where the interplay of ct⁶A tRNA modification levels and TOR pathway activity influence the filamentation of *C. albicans* and *C. dubliniensis* via a common route but with species-specific outcomes. Central in this model is the tRNA-modifying enzyme Hma1, which influences the translation efficiency of many hyphae- and pathogenicity-associated genes. Together with the output of the Tor1-dependent signaling, the combined effect has a strong impact on morphogenesis, invasion, and in consequence virulence of *C. albicans*. To the best of our knowledge, this is the first study to directly link tRNA modifications to fungal pathogenicity as a co-translational factor in morphogenesis.

## ACKNOWLEDGMENTS

We thank Joachim Morschhäuser for providing the genetic library, Rita Müller for help with the *ECE1* transcript analyses, our students and technical assistants for their supporting work, and Bernhard Hube for helpful discussions.

This work was supported in part by the Deutsche Forschungsgemeinschaft [SPP2225 to S.B., STA1147/1-1 to B.B., TRR124 to S.V. and S.A.]; the Bundesministerium für Bildung und Forschung [03Z22JN11 to S.V.]; and the European Research Council [ERC-2012-StG 310489-tRNAmodi to S.A.L.].

## AUTHOR AFFILIATIONS

[1]Department of Microbial Pathogenicity Mechanisms, Leibniz Institute for Natural Product Research and Infection Biology – Hans Knoell Institute, Jena, Germany

[2]Septomics Research Center, Friedrich Schiller University and Leibniz Institute for Natural Product Research and Infection Biology – Hans Knoell Institute, Jena, Germany

[3]Max Planck Research Group for RNA Biology, Max Planck Institute for Molecular Biomedicine, Münster, Germany

[4]Research Group for Cellular RNA Biochemistry, Department of Chemistry, Biochemistry and Pharmaceutical Sciences, University of Bern, Bern, Switzerland

[5]Bioanalytical Mass Spectrometry Unit, Max Planck Institute for Molecular Biomedicine, Münster, Germany

[6]Research Group Microbial Immunology, Leibniz Institute for Natural Product Research and Infection Biology – Hans Knoell Institute, Jena, Germany

[7]Institute of Microbiology, Friedrich Schiller University, Jena, Germany

## AUTHOR ORCIDs

Slavena Vylkova ⓘ http://orcid.org/0000-0002-0051-4664
Sascha Brunke ⓘ http://orcid.org/0000-0001-7740-4570

## FUNDING

| Funder | Grant(s) | Author(s) |
|---|---|---|
| Deutsche Forschungsgemeinschaft (DFG) | SPP2225 | Sascha Brunke |
| Deutsche Forschungsgemeinschaft (DFG) | STA1147/1-1 | Bettina Böttcher |
| Deutsche Forschungsgemeinschaft (DFG) | TRR124 | Slavena Vylkova |
| Bundesministerium für Bildung und Forschung (BMBF) | 03Z22JN11 | Slavena Vylkova |
| EC | European Research Council (ERC) | ERC-2012-StG 310489-tRNAmodi | Sebastian A. Leidel |

## AUTHOR CONTRIBUTIONS

Bettina Böttcher, Conceptualization, Formal analysis, Investigation, Visualization, Writing – original draft, Writing – review and editing | Sandra D. Kienast, Formal analysis, Investigation | Johannes Leufken, Formal analysis, Investigation | Cristian Eggers, Formal analysis, Investigation | Puneet Sharma, Formal analysis, Investigation | Christine M. Leufken, Formal analysis, Investigation | Bianka Morgner, Investigation | Hannes C. A. Drexler, Investigation, Methodology | Daniela Schulz, Investigation | Stefanie Allert, Formal analysis, Investigation, Visualization | Ilse D. Jacobsen, Formal analysis, Investigation | Slavena Vylkova, Conceptualization, Supervision, Writing – review and editing | Sebastian A. Leidel, Conceptualization, Formal analysis, Methodology, Supervision, Writing – review and editing | Sascha Brunke, Conceptualization, Formal analysis, Methodology, Supervision, Visualization, Writing – original draft, Writing – review and editing

## DATA AVAILABILITY

The data sets are available in the following databases: Sequencing data from *Candida albicans* ribosome profiling experiments – Gene Expression Omnibus GSE199421; Data from RNA mass spectrometry – MetaboLights database MTBLS4439.

## ADDITIONAL FILES

The following material is available online.

### Supplemental Material

**Supplemental material (Spectrum04255-22-s0001.docx).** Supplementary Figures S1-S8 and supplementary Tables S1 and S2.

### Open Peer Review

**PEER REVIEW HISTORY (review-history.pdf).** An accounting of the reviewer comments and feedback.

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
