## [Reviewer comments · Microbiology Spectrum]

Microbiology Spectrum

A highly conserved tRNA Modification Contributes to *C. albicans* Filamentation and Virulence

Bettina Böttcher, Sandra Kienast, Johannes Leufken, Cristian Eggers, Puneet Sharma, Christine Leufken, Bianka Morgner, Hannes Drexler, Daniela Schulz, Stefanie Allert, Ilse Jacobsen, Slavena Vylkova, Sebastian Leidel, and Sascha Brunke

Corresponding Author(s): Sascha Brunke, Leibniz Institute for Natural Product Research and Infection Biology - Hans Knöll Institute

Review Timeline:

Submission Date:	October 18, 2022
Editorial Decision:	November 21, 2022
Revision Received:	April 13, 2023
Editorial Decision:	May 8, 2023
Revision Received:	January 12, 2024
Accepted:	January 18, 2024

Editor: Gustavo Goldman

Reviewer(s): Disclosure of reviewer identity is with reference to reviewer comments included in decision letter(s). The following individuals involved in review of your submission have agreed to reveal their identity: Attila Gacser (Reviewer #1); Gary Moran (Reviewer #3)

Transaction Report:

DOI: <https://doi.org/10.1128/spectrum.04255-22>

November 21, 2022

Dr. Sascha Brunke
Leibniz Institute for Natural Product Research and Infection Biology - Hans Knöll Institute (HKI)
Microbial Pathogenicity Mechanisms
Jena
Germany

Re: Spectrum04255-22 (tRNA Modifications Contribute to Fungal Virulence)

Dear Dr. Sascha Brunke:

Your manuscript has been reviewed by two reviewers who suggested several modifications. Please, submit a revised version of the manuscript together with a rebuttal letter addressing point-by-point raised by each reviewer.

Link Not Available

Sincerely,

Gustavo Goldman

Journals Department
Reviewer comments:

Reviewer #1 (Comments for the Author):

The manuscript by Böttcher and coworkers describes the role of the Hma1 (*S. cerevisiae* ortholog TCD2) enzyme in *Candida albicans* and in *Candida dubliniensis*. They show that Hma1 has tRNA-threonylcarbamoyladenine dehydratase activity, and the function of this tRNA-modifying enzyme is species-specific since it influences the hyphal growth differently in the two species. The authors conclude that the tRNA modifications have an important role in the *Candida albicans* pathogenicity and thus could offer a new therapeutic target to combat fungal infections.

The manuscript is well-written, and the results are presented clearly. The figures are informative, and the data support the conclusions. However, I have some questions and suggestions that could help the readers to understand the major message of

the paper better.

The authors suggest that Hma1 has a specific role in TOR-related morphology and nutrient signaling; however, it is also possible that the PKA pathway is also involved and the major downstream targets of this pathway (for example, Efg1, Flo8) can be influenced. Since the Flo8/Efg1 complex regulates several target genes that are important in morphogenesis, it would be informative to show whether or not the regulation of these elements plays any role in the identified phenotypes.

Since intact morphological change is essential in biofilm formation, it could be interesting to test if the mutants are impaired in the biofilm formation capacity.

The authors show the differences in the filamentation of the *C. albicans* and *C. dubliniensis* hma1 knockouts, but only on agar plates. Microscopic images could be helpful to justify the differences.

The authors also show differences in the invasion capacity of the mutants compared to the WT. Was the damaging ability of the mutants also impaired? Since the major candida factor that causes damage in host epithelial cells is the candidalysin, it could be informative to test if the ECE1 expression of the mutants is still intact.

Reviewer #3 (Comments for the Author):

The authors describe a novel tRNA modifying enzyme in the fungal pathogen *C. albicans*, encoded by HMA1. The enzyme plays an important role in regulating TOR kinase activity and many virulence associated traits. The gene was identified in a screen of *C. albicans* genes that could suppress pseudohyphal growth in the related species *C. dubliniensis*. The functions of HMA appear to be divergent in these two species and this indicates that HMA function in *C. albicans* is a recently evolved trait.

Suggestions for comment or additional experiments:

Very little information is given on the homology of the *C. albicans* and *C. dubliniensis* HMA1 genes. Are they similar at the AA sequence level? Are the genes syntenic? Are there any additional orthologous genes in either species?

Are the *C. albicans* and *C. dubliniensis* genes expressed at similar levels? In the original screen, are HMA1 and CdHMA1 both expressed?

Is HMA1 regulated by nutrient source or temperature?

Can any of the filamentation defects in the *C. albicans* hma mutant be rescued by sub-inhibitory concentrations of rapamycin?

Did the investigators attempt to express the *C. dubliniensis* HMA gene in *C. albicans*?

Minor correction: Fig 1C, YBC should be YCB

Staff Comments:

Preparing Revision Guidelines

Please return the manuscript within 60 days; if you cannot complete the modification within this time period, please contact me. If you do not wish to modify the manuscript and prefer to submit it to another journal, please notify me of your decision immediately so that the manuscript may be formally withdrawn from consideration by Microbiology Spectrum.

Response to reviewer comments:

We thank the reviewers and the editor for taking the time to evaluate our manuscript. The reviewers have provided us with valuable suggestions for further tests and clarifications, which we have addressed in this revision. With our additional data and the changes to the text, we think the manuscript has clearly improved, and we hope the reviewers will agree.

Reviewer #1 (Comments for the Author):

The manuscript by Böttcher and coworkers describes the role of the Hma1 (*S. cerevisiae* ortholog TCD2) enzyme in *Candida albicans* and in *Candida dubliniensis*. They show that Hma1 has tRNA-threonylcarbamoyladenine dehydratase activity, and the function of this tRNA-modifying enzyme is species-specific since it influences the hyphal growth differently in the two species. The authors conclude that the tRNA modifications have an important role in the *Candida albicans* pathogenicity and thus could offer a new therapeutic target to combat fungal infections. The manuscript is well-written, and the results are presented clearly. The figures are informative, and the data support the conclusions. However, I have some questions and suggestions that could help the readers to understand the major message of the paper better.

We would like to thank the reviewer for this positive assessment and the constructive suggestions here.

The authors suggest that Hma1 has a specific role in TOR-related morphology and nutrient signaling; however, it is also possible that the PKA pathway is also involved and the major downstream targets of this pathway (for example, Efg1, Flo8) can be influenced. Since the Flo8/Efg1 complex regulates several target genes that are important in morphogenesis, it would be informative to show whether or not the regulation of these elements plays any role in the identified phenotypes.

We cannot exclude the contribution of other pathways, but with the huge overlap of different regulatory networks in hyphae formation, it is difficult to disentangle the TOR and PKA pathways. For example, Tor1 can regulate the activities of Efg1 and Bcr1, but also of the repressor Nrg1 or Tup1, and thereby affect many hyphae-associated genes like *ECE1* and adhesin genes (described e.g. in 1). The contribution of TOR is evident from our data, but for a deeper analysis of other regulators like Efg1, a larger set of combined deletion mutants would be required. We believe this to be beyond the scope of our manuscript, but we have included this interesting open question into the discussion section to make clear that we do not necessarily think that TOR is the only factor in our phenotype.

Since intact morphological change is essential in biofilm formation, it could be interesting to test if the mutants are impaired in the biofilm formation capacity.

As suggested, we tested the biofilm formation capacity of the *HMA1* mutants in comparison to the wild type by XTT assay, and we found no difference between the strains. This is in agreement with our data that shows that on plates and contact with surfaces, filamentation itself is not much reduced, but invasive hyphae seem to be affected. In the biofilm assay, this invasion capability seems not to be needed. We have added this information to the text (on page 8), but decided to not show the data set as a figure (with the absence of any difference).

The data set is shown here, based on two sets of 3 biological replicates with 5 samples each. The biofilms were grown for 3 days and the samples were incubated with XTT for 1h or 3h (the two sets) at 37°C following the protocol of Melo *et al.* (2). XTT reduction was determined by absorbance at 490 nm in a microplate well reader and normalized to the WT for each run. Statistical tests (paired ttest on the absorbance data) showed no significant ($p < 0.05$) differences between the strains.

The authors show the differences in the filamentation of the *C. albicans* and *C. dubliniensis* hma1 knockouts, but only on agar plates. Microscopic images could be helpful to justify the differences.

We have now included micrographs of the hyphae in figure 5 (new panel D), together with the new *ECE1* transcription data, as suggested in the next point.

The authors also show differences in the invasion capacity of the mutants compared to the WT. Was the damaging ability of the mutants also impaired? Since the major candida factor that causes damage in host epithelial cells is the candidalysin, it could be informative to test if the *ECE1* expression of the mutants is still intact.

The reviewer is right, with *ECE1* being the main marker of hyphae length, one would expect a difference in expression levels between the wild type and the *HMA1* deletion mutants. A qPCR test on hyphae grown in liquid medium (water with serum) revealed that to be indeed the case. This new data has been added as the new panel C to figure 5, and is now also described in the text both in the results (page 9) and discussion section (page 13), plus an additional section in the Material and Methods. Overall, *ECE1* transcription is therefore still high in the deletion mutants, but severely reduced compared to the wild type. The data we added in figure 5 is shown also here for context:

Reviewer #3 (Comments for the Author):

The authors describe a novel tRNA modifying enzyme in the fungal pathogen *C. albicans*, encoded by HMA1. The enzyme plays an important role in regulating TOR kinase activity and many virulence associated traits. The gene was identified in a screen of *C. albicans* genes that could suppress pseudohyphal growth in the related species *C. dubliniensis*. The functions of HMA appear to be divergent in these two species and this indicates that HMA function in *C. albicans* is a recently evolved trait.

Suggestions for comment or additional experiments:

Very little information is given on the homology of the *C. albicans* and *C. dubliniensis* HMA1 genes. Are they similar at the AA sequence level? Are the genes syntenic? Are there any additional orthologous genes in either species?

Thanks for this feedback, after working with *C. dubliniensis* for such a long time we have forgotten that it is not necessarily known to all readers that most of its genes are nearly identical to *C. albicans*. We have added the following information on HMA1 to the manuscript: In the new figure panel 3A, we now show the overall structure of ScTcd2 and CaHma1 in direct comparison, with important domains highlighted. For the direct comparison between *C. albicans* and *C. dubliniensis*, we have added to the text (pages 7 & 8) the information that they are near-identical and their genes are syntenic (as is frequently the case between the two species).

The orthologs are not syntenic between *S. cerevisiae* and *C. albicans*, but in our experience, this is rarely the case. A paralog exists in *S. cerevisiae* (also not unusual in this post-WGD yeast), but none in *C. albicans* (or *C. dubliniensis*). We have also added this information to the results section (page 7).

Are the *C. albicans* and *C. dubliniensis* genes expressed at similar levels? In the original screen, are HMA1 and CdHMA1 both expressed?

This is an interesting question. Grumaz *et al.* have analyzed the differential expression of orthologs in *C. albicans* and *C. dubliniensis* (Grumaz *et al.*, 2013 - Suppl. Table 10B). Even in YPD with serum at 37°C, where *C. albicans* filaments and *C. dubliniensis* does not, the normalized transcript levels of the HMA1 orthologs, as determined by RNA-Seq, are virtually identical. This supports our notion that differences in enzyme function, rather than expression, determine the different effects of Hma1. We have added the information on the identical transcript levels to the discussion (page 13). We therefore assume that HMA1 was expressed in the original screen. Based on the well-supported hypothesis that the difference lies in the specific activities, it seems to be a very reasonable assumption, and it would have little effect on the further analysis described in our manuscript.

Is HMA1 regulated by nutrient source or temperature?

We have looked into previously generated data, both from our group and others (e.g. Mundodi *et al.*, 2020). We have not found any evidence for differential regulation of HMA1 on the RNA level, e.g. between 30°C and 37°C. Combined with the previously mentioned data from Grumaz *et al.*, it seems that HMA1 is rather stably transcribed. This does not exclude that HMA1 transcription changes under specific conditions, of course. We have therefore added this interesting question to the discussion (page 13) as something that can be addressed in future analyses.

Can any of the filamentation defects in the *C. albicans* hma mutant be rescued by sub-inhibitory concentrations of rapamycin?

This is an interesting idea, but as we found out, it is not the case. We exposed the strains to different concentrations of rapamycin and determined the hyphal length in water+serum on surfaces and in liquid culture after 4 hours. We used rapamycin concentrations ranging from 1.9 to 30 nM to cover a range of sub-inhibitory concentrations at the low end and compensate for potential binding by serum at the higher end. Overall, the hyphae became shorter rather than longer with increasing concentrations of rapamycin, with no compensation of the phenotype at any concentration. The data for *hma1Δ/Δ* is shown here, but we decided to not include it in the manuscript, as it would require a lengthy explanation and additional material and methods for a negative result that does not bring forward the main message of the manuscript.

Did the investigators attempt to express the *C. dubliniensis* HMA gene in *C. albicans*?

We have not attempted to express the *C. dubliniensis* *HMA1* gene in *C. albicans*. We have screened a library of *C. dubliniensis* genomic fragments in a *C. albicans* background, complementary to the screen described in this manuscript. We did not see a related phenotype in this assay, which may be due to the partially inverse function of the two *HMA1* genes when it comes to filamentation phenotypes. It would be interesting to test this reverse integration in the future, but it would be informative for the biology of *C. dubliniensis*.

Minor correction: Fig 1C, YBC should be YCB

We thank the reviewer for spotting that mistake. It has been corrected in the new figure 2.

1. Bastidas, R.J., Heitman, J. and Cardenas, M.E. (2009) The protein kinase Tor1 regulates adhesin gene expression in *Candida albicans*. *PLoS Pathog*, **5**, e1000294.
2. Melo, A.S., Bizerra, F.C., Freymuller, E., Arthington-Skaggs, B.A. and Colombo, A.L. (2011) Biofilm production and evaluation of antifungal susceptibility amongst clinical *Candida* spp. isolates, including strains of the *Candida parapsilosis* complex. *Med Mycol*, **49**, 253-262.

May 8, 2023

Dr. Sascha Brunke
Leibniz Institute for Natural Product Research and Infection Biology - Hans Knöll Institute
Microbial Pathogenicity Mechanisms
Jena
Germany

Re: Spectrum04255-22R1 (tRNA Modifications Contribute to Fungal Virulence)

Dear Dr. Sascha Brunke:

Your manuscript has been revised by two reviewers who suggested some modifications. Please, submit a revised version together with a rebuttal letter addressing point-by-point raised by each reviewer.

Link Not Available

Sincerely,

Gustavo Goldman

Journals Department
Reviewer comments:

Reviewer #3 (Comments for the Author):

Thank you for the clarifications, responses and extra work.

Reviewer #4 (Comments for the Author):

In this well conducted study the authors identify Hma1, a tRNA modifying enzyme in *Candida albicans* that is important for the hyphal transition and full virulence of this pathogen. This is interesting as not much is known with respect to the roles of tRNA

modifications in fungal pathogens. The authors describe different roles of Hma1 for filamentation in *C. albicans* and *C. dubliniensis*, which they discuss may relate to their different pathogenicity. However, this last part is in my opinion sufficiently investigated, leaving the comparative aspect of the study unfinished

Given the strategy utilised to identify Hma1, it is expectable that the *albicans* and the *dubliniensis* enzymes have different molecular functions, but this has not been proven. The authors should include the *dubliniensis* *hma1* Δ/Δ in the RNA mass spectrometry experiment (figure 3C) to see if the t6A levels also change in this species.

This would solve a relevant open question, does Hma1 have different functions in *C. albicans* and *C. dubliniensis* because the activity is not the same, or because the role of t6A is different?

Does expression of *C. albicans* Hma1 in *C. dubliniensis* (strain Cd2115) induce/enhance filamentation? If so, does this make *C. dubliniensis* invasive?

- In my opinion the title is too generic and can mislead to the reader. This work only looks at one tRNA modification, and describes its importance for hyphal morphology and virulence (this last just in *C. albicans*)
- Line 337, please, include the alignment in a supplementary figure. It would be interesting to see the few differences between the proteins.
- Figure 2B. If I am not confused, rapamycin is a mTOR antagonist, not agonist. *C. dubliniensis* is noticeably more sensitive to rapamycin than *C. albicans*. Was this known? Potential explanations, including a role for Hma1? It would be interesting to discuss this point.
- Figure 2C. The differences with Homoserine are very subtle, and I wonder if biologically relevant. The authors state that the *C. albicans* *hma1* Δ/Δ grew slightly better than wt, but that is minor, and varies between clones. Next, the authors state that in *C. dubliniensis* the mutant grew as the wt, but I see one of the clones growing better, and the two red lines over the black one most of the time. I would suggest that the authors assay a higher concentration of homoserine to maximise the effect and clarify if there are real differences.
- Line 388-390. I do not understand this result. Where are these 77 samples coming from? Can the authors include exemplary individual spectra to show how ct6A37 is or is not detected?
- Lines 406-410 (figure S6). I am confused here, the authors state that *C. dubliniensis* does not form hyphae on standard agar plates, but are there not hyphae in the plates in fig S6? The quality of these photos is not good, which complicates the evaluation.
- Figure 6. Can the authors show some primary data here (i.e. microscopic photos)?
- Figure 7. It is not fair to only show the mutant with a more pronounced decrease in virulence in the main figure, and the other clone only in a supplementary. Please, include both in the main figure. On which mutant clone was the reverted strain constructed? The one with highest or lower decrease in virulence? This can be discussed.

Staff Comments:

Preparing Revision Guidelines

Please return the manuscript within 60 days; if you cannot complete the modification within this time period, please contact me. If you do not wish to modify the manuscript and prefer to submit it to another journal, please notify me of your decision immediately so that the manuscript may be formally withdrawn from consideration by Microbiology Spectrum.

Point-by-point response to the reviewer comments

In this well conducted study the authors identify Hma1, a tRNA modifying enzyme in *Candida albicans* that is important for the hyphal transition and full virulence of this pathogen. This is interesting as not much is known with respect to the roles of tRNA modifications in fungal pathogens. The authors describe different roles of Hma1 for filamentation in *C. albicans* and *C. dubliniensis*, which they discuss may relate to their different pathogenicity. However, this last part is in my opinion sufficiently investigated, leaving the comparative aspect of the study unfinished.

We thank the reviewer for this positive assessment of our work and the constructive criticism here and in the following questions. We somewhat disagree with the focus on the comparative aspect, as we will try to make clear in our answers to the individual points raised by the reviewer. To us, the direct comparison is the starting point of this investigation, and many of our conclusions are based on the differences in the phenotypes of the *C. albicans* and *C. dubliniensis* HMA1 deletion mutants. Indeed, as outlined below, direct comparison experiments are often prohibitively difficult and complex. With the information we provide below and with the changes we made to the manuscript based on the questions, we are confident that we provide sufficient arguments for our approach.

Given the strategy utilised to identify Hma1, it is expectable that the *albicans* and the *dubliniensis* enzymes have different molecular functions, but this has not been proven. The authors should include the *dubliniensis* *hma1Δ/Δ* in the RNA mass spectrometry experiment (figure 3C) to see if the t⁶A levels also change in this species.

While we agree that this may be an interesting experiment for future work, we don't think this to be strictly necessary for this manuscript. Our main claim is that the *C. albicans* Hma1 enzyme plays important roles in its hyphae formation and virulence, and we think our data supports that quite well. Given the time-consuming and complicated nature of the specialized MS experiments, we believe such an investigation would be more suited for a manuscript that specifically deals with *C. dubliniensis*. We see where the reviewer's idea comes from, and we agree that it is of interest for future research into *C. dubliniensis* pathobiology. Therefore, we have added the following sentence to the discussion:

"Whether this is due to intrinsic differences in the enzymatic activity between the two orthologues, different expression pattern or the underlying biological differences between the two species, is an interesting open question for future research."

This would solve a relevant open question, does Hma1 have different functions in *C. albicans* and *C. dubliniensis* because the activity is not the same, or because the role of t⁶A is different? Does expression of *C. albicans* Hma1 in *C. dubliniensis* (strain Cd2115) induce/enhance filamentation? If so, does this make *C. dubliniensis* invasive?

The filamentation of strain Cd2115 (*C. dubliniensis* containing a *C. albicans* HMA1 gene) does not differ from its parental strain (Wü284) — this is shown in supplementary figure 7. We have also tested the invasiveness of Cd2115 by measuring LDH release from human epithelial cells, and found no difference to the Wü284 wild type. A general problem with these experiments is that HMA1 often seems to have opposite effects in the two species. Deletion of *CdHMA1*, for example, leads to longer hyphae — and presence of HMA1 in *C. albicans* is also associated with longer hyphae there. Therefore, it would be difficult to see the effect of expression of (*C. albicans*) HMA in a *CdHMA1*-deleted *C. dubliniensis* background (the case of Cd2115). If *CdHMA1* is not deleted, gene dosage effects may confound the interpretation of the data. Overall, we therefore believe that to reliably answer all these further questions, a large set of new experiments is required, which we consider to be beyond the scope of

our current manuscript. As mentioned above, we now put this question before the readers in the discussion section, also to show the limits of the present study.

- In my opinion the title is too generic and can mislead to the reader. This work only looks at one tRNA modification, and describes its importance for hyphal morphology and virulence (this last just in *C. albicans*)

To address these concerns, we have changed the title to “A highly conserved tRNA Modification Contributes to *C. albicans* Filamentation and Virulence”.

- Line 337, please, include the alignment in a supplementary figure. It would be interesting to see the few differences between the proteins.

We have added the alignment as the new figure S2.

- Figure 2B. If I am not confused, rapamycin is a mTOR antagonist, not agonist. *C. dubliniensis* is noticeably more sensitive to rapamycin than *C. albicans*. Was this known? Potential explanations, including a role for Hma1? It would be interesting to discuss this point.

We thank the reviewer for spotting that mistake, and we have corrected the label of figure 2B. Concerning the sensitivity of *C. dubliniensis* to rapamycin, there doesn't seem to be a systematic investigation into this phenomenon. However, striking differences in TOR signaling have been described before by Moran *et al.*, which we now cite as a relevant observation that may help in understanding the effect. We have added the following section to the discussion:

“Generally, *C. dubliniensis* was more sensitive to rapamycin in our growth assays. The reason is not clear, but we note that differences in TOR activation have been linked to the lower rate of filamentation in *C. dubliniensis* compared to *C. albicans* [Moran, 2012]. Potentially, the basal TOR activity level or the downstream signaling – including modulation by Hma1 – may differ between the two fungi [Moran, 2012], making *C. dubliniensis* more sensitive to perturbations.”

- Figure 2C. The differences with Homoserine are very subtle, and I wonder if biologically relevant. The authors state that the *C. albicans hma1Δ/Δ* grew slightly better than wt, but that is minor, and varies between clones. Next, the authors state that in *C. dubliniensis* the mutant grew as the wt, but I see one of the clones growing better, and the two red lines over the black one most of the time. I would suggest that the authors assay a higher concentration of homoserine to maximise the effect and clarify if there are real differences.

We repeated the growth curve experiments with higher concentrations of homoserine (4 mg/ml) as suggested, and we do indeed see a more distinct phenotype. As observed in other experiments, the revertant is somewhere intermediate between mutant and wild type (likely due to dose-dependent effects), and the *C. albicans* (A and B) mutants are clearly different to the wild type. For *C. dubliniensis*, where the wild type grows better than *C. albicans* already, the mutants do not lead to any measurable difference in the growth behavior. We have replaced the graph in figure 2C with the new data and changed the text to reflect the higher concentration that is now used. We would like to thank the reviewer for this valuable suggestion.

- Line 388-390. I do not understand this result. Where are these 77 samples coming from? Can the authors include exemplary individual spectra to show how ct⁶A is or is not detected?

We have added a more detailed explanation of our method to determine the presence of ct⁶A, and the technical limitations that precluded us from determining it more directly. This can now be found in the material and methods section:

“pyQms calculates an accurate isotope pattern for each molecule and generates a quality score for each MS1 measurement. Therefore, it does not only use the m/z value of a given molecule but extracts the chemical sum formula of each ion. The ct⁶A signal in the biological measurements was so low that the mass spectrometer did not select the ion for MS2 fragmentation. Therefore, we report the detection of a weak ct⁶A signal in cases where pyQms detected an ion with the sum formula of ct⁶A (C₁₅H₁₈N₆O₇) and a high pyQms quality score at the same retention time as the ct⁶A chemical standard.”

This is an example of the data we obtained with this method (MORAST/pyQms screenshots):

Left: Detection of ct⁶A standard as described above and in the new Material and Methods section

Below: Signal detected in a biological sample, here considered positive.

We also have changed the text in the results section to better explain the type of data we obtained, and we now also provide an overview of the samples in the new Supplementary Table S2. The text now reads: “The total amount of ct⁶A was insufficient to obtain quantitative results due to its limited stability. However, we detected a weak MS1 signal indicating the presence of ct⁶A in 75 of 77 samples (49/50

[98%] in *C. albicans* and 27/27 [100%] in *C. dubliniensis*, over all biological and technical replicates and conditions; Supp. Table 2) from the wild types and the revertant, but only in 2 of 50 samples (4%) taken from the deletion mutants. This signal was observed at the same retention time as a chemically synthesized nucleoside standard for ct⁶A.”

We have also added a further explanation to the Material and Methods section: “For each biological replicate we prepared 2-3 technical replicates depending on the amount of tRNA that we extracted from the gel (50 samples in total for *C. albicans* wt and revertant, 50 for *C. albicans* deletion mutants, and 27 for *C. dubliniensis* wt; Supp. Table S2).”

The new Supplementary Table S2 with the overview of all samples (as runs with positive ct6A signals) is reproduced below. We would also like to stress again that these data are there to provide additional support, from the edge of technical possibilities, to the actual detection of the educt in figure 3.

	Total Runs	ct ⁶ A Signal	Percent Positive
C. albicans			
WT SC5314 YPD 30°C	10	10	100%
Revertant YPD 30°C	9	9	100%
WT SC5314 YPD 37°C	9	9	100%
Revertant YPD 37°C	9	9	100%
WT SC5314 Serum	6	6	100%
Revertant Serum	7	6	86%
Total	50	49	98%
C. albicans deletion strains			
2115A YPD 30°C	10	0	0%
2115B YPD 30°C	8	0	0%
2115A YPD 37°C	9	0	0%
2115B YPD 37°C	9	0	0%
2115A Serum	8	2	25%
2115B Serum	6	0	0%
Total	50	2	4%
C. dubliniensis			
Wue284 YPD 30°C	10	10	100%
Wue284 YPD 37°C	9	9	100%
Wue284 Serum	8	8	100%
Total	27	27	100%

- Lines 406-410 (figure S6). I am confused here, the authors state that *C. dubliniensis* does not form hyphae on standard agar plates, but are there not hyphae in the plates in fig S6? The quality of these photos is not good, which complicates the evaluation.

The plates shown in figure S6 (now S7) show what the text describes as “a late robust hyphae formation without central wrinkling” on minimal medium (here, SLAD, Spider, water after 14 days). To avoid any possible confusion, we have moved the reference to figure (now) S7 directly behind the statement about the late filamentation on minimal media.

- Figure 6. Can the authors show some primary data here (i.e. microscopic photos)?

We are reluctant to add more micrographs here, as – in our opinion - they won't provide any benefit to the reader. The pictures that were taken for large-scale counting/measuring with well-established methods, and are very cluttered for use as a figure. The hyphae are generally similar to the ones shown in figure 5.

- Figure 7. It is not fair to only show the mutant with a more pronounced decrease in virulence in the main figure, and the other clone only in a supplementary. Please, include both in the main figure. On which mutant clone was the reverted strain constructed? The one with highest or lower decrease in virulence? This can be discussed.

We have shown the mutant for which we have created the revertant in figure 7 (*hma1Δ/Δ-B*, as can be seen in Table 1). Following the reviewer's suggestion, we have now included the data from *hma1Δ/Δ-A* and changed the figure legend accordingly. The new text also states more clearly that the revertant is based on the *hma1Δ/Δ-B* mutant, to which it was directly compared in virulence.

Re: Spectrum04255-22R2 (A highly conserved tRNA Modification Contributes to *C. albicans* Filamentation and Virulenc)

Dear Dr. Sascha Brunke:

Your manuscript is now ready for publication. Congratulations !!!!!

Your manuscript has been accepted, and I am forwarding it to the ASM production staff for publication. Your paper will first be checked to make sure all elements meet the technical requirements. ASM staff will contact you if anything needs to be revised before copyediting and production can begin. Otherwise, you will be notified when your proofs are ready to be viewed.

Sincerely,
Gustavo H. Goldman
Editor
Microbiology Spectrum

Reviewer #4 (Comments for the Author):

Thank you for your clear responses.

I would only like to mention that showing primary data (i.e. the microscopy photos) is always valuable for the reader and can provide additional information and support to the results. If the main figure is too busy, it can be shown as a supplementary.

Nevertheless, I am not doubting the authors' results. I am satisfied with the responses.